# CD8⁺ T cell stemness precedes post-intervention control of HIV viraemia

Zahra Kiani[1,11], Jonathan M. Urbach[1,11], Hannah Wisner[1,11], Mpho J. Olatotse[1], Daniel Y. Chang[1,2], Joshua A. Acklin[1], Alicja Piechocka-Trocha[1,3], Nathalie Bonheur[1], Ashok Khatri[4], Mathias Lichterfeld[1,5], Jesper D. Gunst[6,7], Ole S. Søgaard[6,7], Marina Caskey[8], Michel C. Nussenzweig[3,8], Bruce D. Walker[1,3,9,10] & David R. Collins[1,3 ✉]

Interventions to induce lasting human immunodeficiency virus (HIV) remission are needed to obviate the requirement for lifelong antiretroviral therapy. Durable post-intervention control (PIC) of viraemia has been achieved in a subset of people following administration of broadly neutralizing anti-HIV-1 antibodies (bNAb) and analytical interruption of treatment[1–4]. Previous studies support a role for CD8⁺ T cells in PIC[5–9], but the precise features of CD8⁺ T cells involved remain unclear. Here we mapped and functionally profiled CD8⁺ T cell responses to autologous HIV epitopes using longitudinal samples from four analytical treatment interruption trials in bNAb recipients. PIC was associated with superior pre-intervention HIV-specific CD8⁺ T cell proliferative capacity, stem-cell-like memory phenotype and recall cytotoxicity against autologous HIV peptide-pulsed CD4⁺ T cells. CD8⁺ T cell stemness was increased further following bNAb administration without emergence of new clonotypes targeting defined HLA-optimal epitopes. Multi-modal single-cell analyses revealed molecular features associated with PIC and HIV-specific CD8⁺ T cell stemness, including signatures of metabolic fitness and reduced T cell exhaustion. These results identify immune features that precede subsequent PIC to inform the development of combination immunotherapies that will elicit durable HIV remission.

Approximately 40 million people worldwide are living with HIV, requiring lifelong antiretroviral therapy (ART) to prevent recrudescent viral replication, transmission and disease progression[10]. To inform the development of a functional cure by which durable ART-free remission can be achieved, mechanisms underlying spontaneous control of HIV to undetectable levels without ART have been studied extensively[11,12]. The proliferative capacity of HIV-specific memory CD8⁺ T cells has been linked repeatedly to spontaneous control[13–15], is associated with increased stemness[16] and facilitates lytic granule loading for cytotoxic elimination of HIV-infected cells[17]. Moreover, loss of these functions precedes aborted spontaneous control of HIV[18].

A small fraction of people living with HIV can maintain low or undetectable plasma viral loads for a variable period following discontinuation of ART[19,20]. Specifically, 4% of participants in non-interventional analytical treatment interruption (ATI) trials achieved control of viraemia for 84 days or more[21]. Such post-treatment control (PTC) has been associated with particular virologic and immunologic characteristics, such as smaller persistent HIV reservoirs, autologous virus neutralization and reduced T cell activation[22–24], although precise determinants remain under investigation. Efforts to achieve durable post-ART control in a larger proportion of people living with HIV have combined ATI with interventions such as passive bNAb infusion[1–4,25]. Although

PIC of viraemia has been achieved following bNAb administration at higher rates than PTC in non-interventional trials, most bNAb recipients did not durably control viraemia, highlighting the need for a deeper understanding of immune responses that mediate PIC[26]. Control of viraemia following bNAb administration in non-human primates was lost upon depletion of CD8⁺ T cells[5–7], demonstrating their importance in PIC. Although modest augmentation of virus-specific CD8⁺ T cells has been observed in vivo following bNAb administration[5–9], the precise CD8⁺ T cell features and functions associated with PIC and the extent to which their augmentation facilitates PIC remain unclear.

Here we identify immune correlates preceding subsequent PIC by studying CD8⁺ T cell responses targeting autologous HIV epitopes in longitudinal specimens obtained from participants in four similar interventional trials. PIC was not associated with broadening of HIV-specific responses against autologous HLA-optimal epitopes following bNAb administration but was associated significantly with superior pre-intervention proliferative and cytolytic potential of HIV-specific stem-cell-like memory CD8⁺ T cells. These responses were further enhanced following bNAb administration and were associated with changes in metabolic gene expression. These immune correlates of PIC may inform strategies to elicit ART-free control of viraemia in a larger proportion of people living with HIV.

[1]Ragon Institute of Mass General Brigham, MIT and Harvard, Cambridge, MA, USA. [2]Department of Pathology, Mass General Brigham, Boston, MA, USA. [3]Howard Hughes Medical Institute, Chevy Chase, MD, USA. [4]Mass General Brigham Peptide Research Core, Massachusetts General Hospital, Charlestown, MA, USA. [5]Division of Infectious Diseases, Brigham and Women's Hospital, Boston, MA, USA. [6]Department of Infectious Diseases, Aarhus University Hospital, Aarhus, Denmark. [7]Department of Clinical Medicine, Aarhus University, Aarhus, Denmark. [8]Laboratory of Molecular Immunology, The Rockefeller University, New York, NY, USA. [9]Institute for Medical Engineering and Science, Massachusetts Institute of Technology, Cambridge, MA, USA. [10]Department of Biology, Massachusetts Institute of Technology, Cambridge, MA, USA. [11]These authors contributed equally: Zahra Kiani, Jonathan M. Urbach, Hannah Wisner. ✉e-mail: drcollins@mgh.harvard.edu

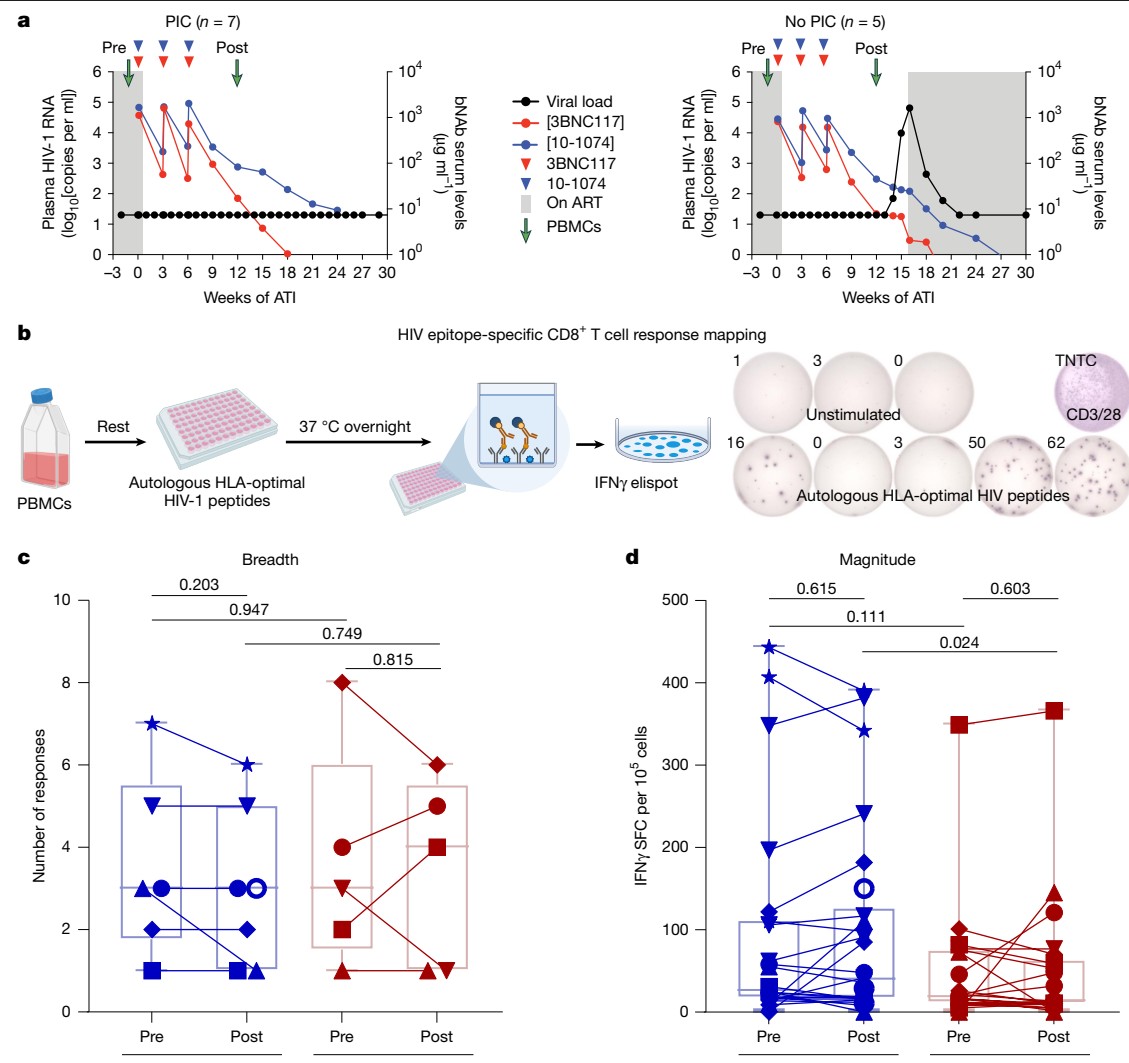

**Fig. 1 | Autologous HIV epitope-specific CD8+ T cell responses in post-intervention controllers. a**, Study cohort overview. Longitudinal PBMC samples were used from seven controllers and five non-controllers before and after infusion of bNAbs 3BNC117 and 10-1074 in the MCA-906, MCA-965, TITAN and eCLEAR trials. **b**, Schematic overview of autologous HIV-specific CD8+ T cell response mapping and representative IFNγ ELISpot results. **c,d**, Summary of longitudinal and between-group differences in breadth ($n = 6, 7, 5, 5$ samples (**c**)) and magnitude ($n = 23, 26, 22, 22$ responses (**d**)) of HIV epitope-specific

responses. SFC, spot-forming counts. Centre lines represent medians, boxes represent first and third quartiles and whiskers represent ranges. Symbols represent individual participants within the PIC and non-PIC groups (key in Table 1). $P$ values reported above plots from two-sided paired (longitudinal) or unpaired (between-group) $t$-tests. Representative diagrams in **a** were modified from ref. 1, Spinger Nature Limited. Schematic in **b** was created using BioRender (https://biorender.com).

## PIC is not associated with broadening of responses

We obtained longitudinal samples of peripheral blood mononuclear cells (PBMCs) before and after passive infusion of bNAbs 3BNC117 and/or 10-1074 in 12 participants from four ATI trials[1–4]: seven individuals who achieved PIC ('controllers') and five who did not ('non-controllers') (Fig. 1a and Table 1). We synthesized peptides matching class I HLA-optimal HIV epitopes encoded by autologous intact proviral DNA sequenced from each participant[27] (Supplementary Data 1) and mapped epitope-specific CD8+ T cell responses by interferon-γ enzyme-linked immunospot (IFNγ ELISpot; Fig. 1b). A mean of 3.5 (range 1–8) HIV epitope-specific CD8+ T cell responses per participant was identified, and neither response breadth, induction of new responses against HLA-optimal HIV epitopes reported to be presented by the expressed HLA class I alleles nor magnitude of IFNγ production was associated with PIC (Fig. 1c,d and Supplementary Data 1). These data indicate that the induction of de novo CD8+ T cell responses against known HLA-optimal HIV epitopes following bNAb administration is not a unique correlate of PIC.

## HIV-specific CD8+ T cell stemness precedes PIC

Because proliferation is better correlated with cytotoxic function and spontaneous control of HIV viraemia than IFNγ production[13,14,16–18], we next measured the ability of CD8+ T cells to proliferate upon stimulation with cognate HIV peptides corresponding to each response identified by IFNγ ELISpot (Fig. 2a,b and Extended Data Fig. 1a). Pre-intervention proliferative capacity of CD8+ T cells against autologous HIV epitopes was on average more than tenfold higher in controller responses relative to non-controller responses (mean 9.7% versus 0.9%, median 3.6% versus 0.3% CFSE-low; $P < 0.001$; Fig. 2c). Notably, participant 314 had especially strong proliferative responses (range 16.0–49.7% CFSE-low) against five distinct epitopes (Fig. 2c) and was the only participant whose intact HIV DNA reservoir was below the assay detection limit (Table 1), suggesting a potential role for highly functional HIV epitope-specific CD8+ T cells in limiting HIV persistence in this participant. Proliferative capacity remained significantly higher in controller responses than non-controller responses even when responses from this participant were excluded from analysis. Following bNAb administration, the proliferative capacity

## Table 1 | Participant clinical and demographic characteristics

| PID | Study | Intervention | Phenotype | Samples | Age (years) | Sex | Race | HIV/ART duration | CD4 count | Intact HIV DNA | HLA | HIV epitopes |
|---|---|---|---|---|---|---|---|---|---|---|---|---|
| 9243 ▼ | MCA-906 | 3× 3BNC117+10-1074 during ATI | No PIC | ATI start; 12 w post-ATI | 29 | M | AI, H | 5 y/5 y | 583 | 0.17 | A24,30 B15,31 C02,15 | 49 A |
| 9252 ▲ | MCA-906 | 3× 3BNC117+10-1074 during ATI | No PIC | ATI start; 12 w post-ATI | 51 | F | B | 11 y/11 y | 598 | 1.71 | A02,66 B39,78 C12,16 | 37 A |
| 9254 ♦ | MCA-906 | 3× 3BNC117+10-1074 during ATI | PIC | ATI start; 12 w post-ATI | 48 | M | W | 21 y/21 y | 860 | NA | A01,29 B38,44 C12,16 | 22 A |
| 9255 ▲ | MCA-906 | 3× 3BNC117+10-1074 during ATI | PIC | ATI start; 12 w post-ATI | 30 | M | W | 5 y/4 y | 1,360 | 1.89 | A03,25 B18,44 C07,12 | 41 A |
| 5106 ■ | MCA-965 | 7× 3BNC117+10-1074 during ATI | PIC | ATI start; 12 w post-ATI | 31 | M | B | 6 y/6 y | 671 | 7.3 | A03,03 B18,<u>57</u> C12,18 | 61 A |
| 5111 ■ | MCA-965 | 7× 3BNC117+10-1074 during ATI | No PIC | ATI start; 12 w post-ATI | 55 | M | W | 20 y/16 y | 760 | 2.5 | A11,32 B35,44 C05,12 | 38 A |
| 5114 ● | MCA-965 | 7× 3BNC117+10-1074 during ATI | No PIC | ATI start; 12 w post-ATI | 54 | M | B | 15 y/15 y | 545 | 6.1 | A03,68 B07,15 C07,07 | 53 A |
| 5120 ● | MCA-965 | 7× 3BNC117+10-1074 during ATI | PIC | ATI start; 12 w post-ATI | 50 | M | W | 19 y/19 y | 1,189 | 0.8 | A02,29 <u>B14</u>,44 C01,03 | 42 A |
| 107 ○ | eCLEAR | 2× 3BNC117+3× RMD at ART initiation | PIC | Post-bNAb (pre-ATI) | 45 | M | W | 1.2 y/1 y | 650 | 50.2 | A02,25 B15,44 C03,05 | 43 A |
| 109 ♦ | TITAN | 2× 3BNC117+10-1074 during ATI | No PIC | ATI start; 6 w post-ATI | 57 | M | W | 5 y/5 y | 1,250 | 93.5 | A02,02 B07,51 C04,07 | 53 C |
| 142 ★ | TITAN | 2× 3BNC117+10-1074 during ATI | PIC | ATI start; 6 w post-ATI | 57 | M | W | 5 y/5 y | 1,210 | 220.2 | A01,02 B08,44 C05,07 | 48 A |
| 314 ▼ | TITAN | 2× 3BNC117+10-1074 during ATI | PIC | ATI start; 6 w post-ATI | 55 | F | W | 2 y/2 y | 1,030 | <1.2 | A30,32 B13,51 C06,14 | 31 C |

Columns show: PID and symbol identifying each participant in the PIC and no PIC groups; parent study; intervention; phenotype; longitudinal sampling; age; biological sex; race and ethnicity; duration of HIV infection and ART before ATI; pre-intervention CD4 count (cells per µm$^3$ peripheral blood); intact HIV per 10$^6$ PBMCs reported previously as measured by qualitative and quantitative viral outgrowth assay (infectious units per million cells, MCA-906 (ref. 1), quadruplex PCR (MCA-965 (ref. 2)) or intact proviral DNA assay (eCLEAR[3], TITAN[4]); class I HLA alleles (protective alleles underlined); and total number of HLA-optimal HIV epitopes screened (A, autologous; C, clade B consensus). Plasma viral loads were undetectable (less than 20 HIV RNA copies ml$^{-1}$) at all sample time points. AI, American Indian; B, Black; F, female; H, Hispanic; M, male; NA, not available; NR, not reported; PID, participant ID; RMD, romidepsin; w, weeks; W, white; y, years.

of responses from both participant groups increased modestly but significantly (median 1.3-fold; $P < 0.01$ controllers, $P < 0.05$ non-controllers; Fig. 2c) and remained significantly higher in controllers than non-controllers following intervention (mean 10.6% versus 1.3%, median 3.8% versus 0.4% CFSE-low; $P < 0.001$; Fig. 2c). This modest increase was consistent with previous observations attributed to a potential bNAb-induced vaccinal effect[5-9] but was not unique to participants who controlled viraemia. Instead, control of viraemia was associated with HIV epitope-specific CD8$^+$ T cell proliferative capacities that were higher before and further enhanced following intervention.

To assess the ability of HIV epitope-specific CD8$^+$ T cells to mount cytotoxic recall responses against autologous CD4$^+$ T cells pulsed with cognate HIV peptides, we performed expanded antigen-specific elimination assays[28] on immunodominant responses from participants with sufficient specimen availability (Fig. 2d,e and Extended Data Fig. 1b). Recall cytotoxicity was associated strongly with proliferative capacity (Spearman $\rho = 0.80$, $P < 0.0001$; Fig. 2f), consistent with previous data from spontaneous HIV controllers[17,18,28] and further supporting a role for highly functional HIV-specific CD8$^+$ T cells in PIC.

To further characterize functional HIV-specific CD8$^+$ T cells in PIC, we next assessed their ex vivo phenotypes by measuring surface expression of differentiation markers on unstimulated peptide-HLA (pHLA)

multimer-stained CD8$^+$ T cells (Fig. 2g, Extended Data Fig. 1c and Supplementary Data 1). HIV epitope-specific CD8$^+$ T cells in controllers had a higher proportion of CD45RA$^+$CD62L$^+$ stem-cell-like memory cells (T$_{SCM}$, $P < 0.05$; Fig. 2h), whereas those from non-controllers had a higher proportion of CD45RA$^-$CD62L$^-$ effector-memory (T$_{EM}$) cells before intervention ($P < 0.01$; Fig. 2i). In comparison, CD8$^+$ T cell responses to cytomegalovirus (CMV) or influenza virus had a higher proportion of CD45RA$^+$CD62L$^-$ terminally differentiated T$_{EMRA}$ cells (Extended Data Fig. 1d). The frequency of T$_{SCM}$ cells among HIV epitope-specific CD8$^+$ T cells in controllers modestly (median 1.2-fold) but significantly ($P < 0.01$) increased following bNAb administration and remained significantly higher post-intervention than in non-controllers ($P < 0.01$; Fig. 2h). Moreover, T$_{SCM}$ frequency was proportional to proliferative capacity (Spearman $\rho = 0.64$, $P < 0.01$; Fig. 2j). Together, these results implicate HIV epitope-specific CD8$^+$ T cell stemness in PIC.

## Molecular signatures of CD8$^+$ T cell stemness in PIC

To identify molecular signatures underlying the superior functional capacity of HIV-specific CD8$^+$ T cells in controllers, we next assessed differential expression of genes and surface proteins among HIV and CMV epitope-specific CD8$^+$ T cells at single-cell resolution by cellular

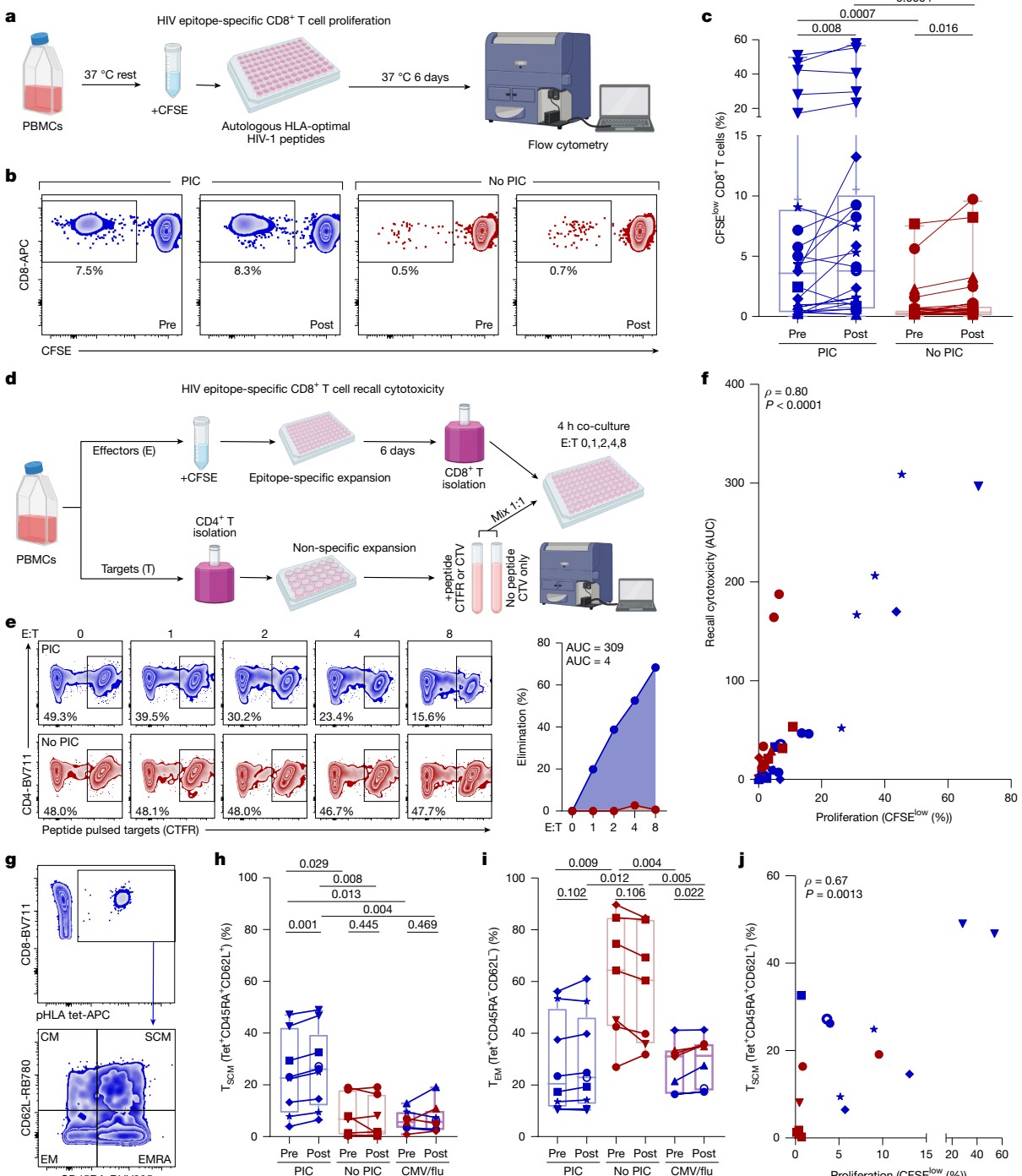

**Fig. 2 | HIV-specific CD8⁺ T cell stemness precedes PIC. a**,**b**, Schematic overview of HIV-specific CD8⁺ T cell proliferation assay (**a**) and representative longitudinal epitope-specific proliferation from one controller (PID 5120) and one non-controller (PID 9243 (**b**)). **c**, Summary of longitudinal and between-group differences in proliferative capacity of CD8⁺ T cell responses against each autologous HIV-1 epitope for which responses were detected by IFNγ ELISpot. Each data point represents the mean of triplicate wells for each response (*n* = 23, 26, 22, 22 responses). **d**,**e**, Schematic overview of expanded antigen-specific elimination assay to measure recall cytotoxicity (**d**) and representative results at increasing effector to target (E:T) ratios from one controller (PID 142; blue) and one non-controller (PID 109; red), including area under the curve (AUC) summaries (**e**). **f**, Correlation of proliferation and recall cytotoxicity, as measured in **d** and **e**, across responses from both pre- and post-intervention samples in controllers (blue) and non-controllers (red). Correlation (*ρ*) and *P* values calculated by Spearman correlation (*n* = 41 responses). **g**, Representative flow cytometric staining of memory subset markers CD45RA

and CD62L on HIV peptide-HLA (pHLA) tetramer⁺ CD8⁺ T cells. **h**,**i**, Summary of longitudinal and between-group differences in T$_{SCM}$ (**h**) and T$_{EM}$ (**i**) subset frequencies among HIV pHLA tetramer⁺ (Tet⁺) CD8⁺ T cell responses from controllers (*n* = 9 responses) and non-controllers (*n* = 7 responses), and among CMV/flu Tet⁺ CD8⁺ T cells from both groups (*n* = 8). **j**, Correlation (*ρ*) and *P* values calculated by Spearman correlation between proliferative capacity and percent T$_{SCM}$ cells among Tet⁺ CD8⁺ T cells in controllers (blue) and non-controllers (red), *n* = 16 responses. Centre lines represent medians, ticks represent means, boxes represent first and third quartiles and whiskers represent ranges. Symbols represent individual participants in PIC and no PIC groups (key in Table 1). *P* values reported above plots from two-sided Wilcoxon signed rank (between-group) or matched-pairs signed rank (longitudinal) tests (**c**), two-sided unpaired (between-group) or paired (longitudinal) *t*-tests (**h**,**i**) or Spearman correlation tests (**f**,**j**). Schematics in **a** and **d** were created using BioRender (https://biorender.com).

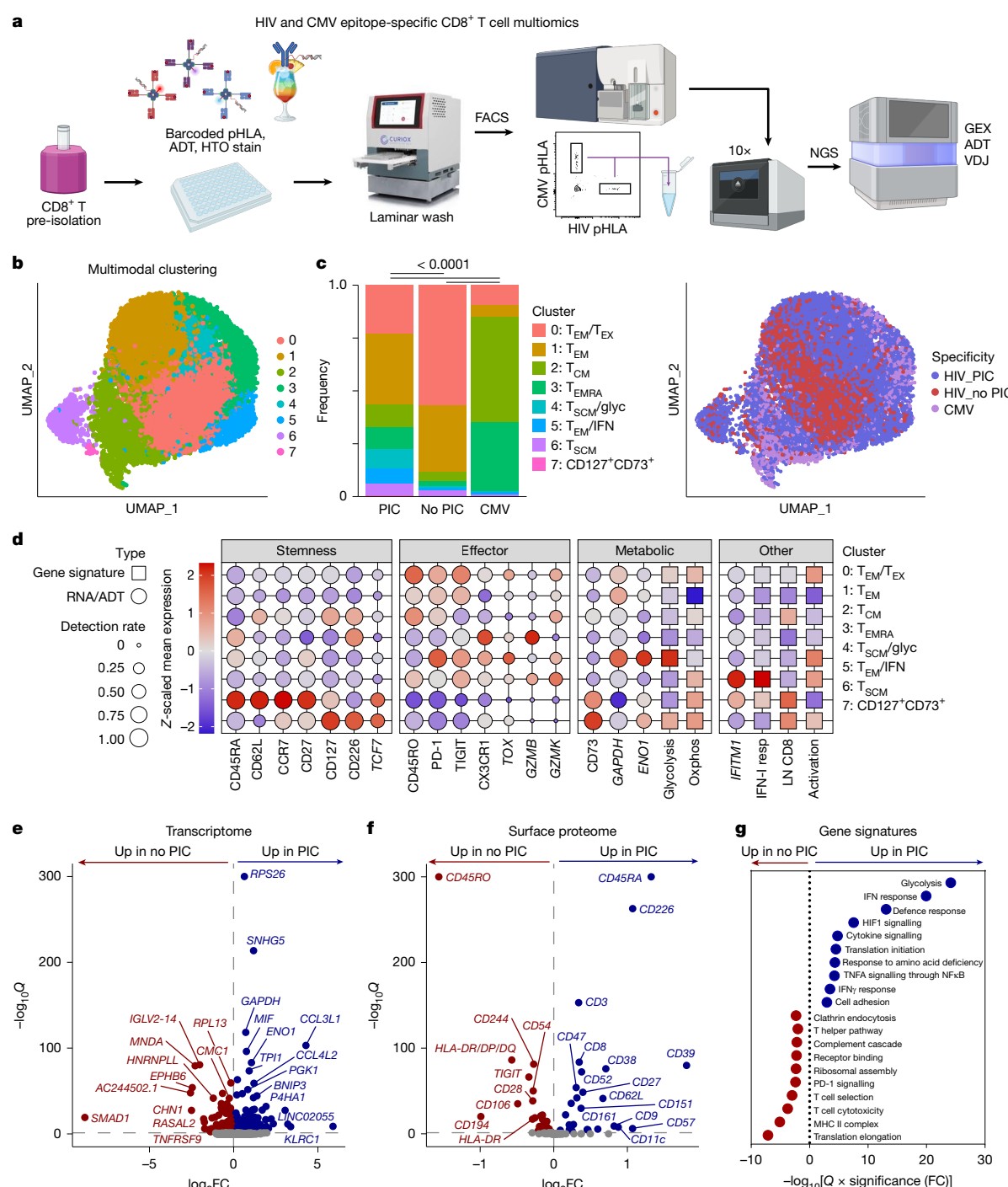

**Fig. 3 | Molecular signatures associated with PIC. a**, Schematic overview of processing, isolation and multiomics sequencing of HIV and CMV epitope-specific CD8+ T cells. **b**, Multi-modal clustering by weighted nearest-neighbours plotted using uniform manifold approximation and projection (UMAP) for dimension reduction. **c**, Left, cluster frequencies among HIV-specific CD8+ T cells from both pre- and post-intervention samples in controllers and non-controllers and among CMV-specific CD8+ T cells and with cluster annotations based on differential expression of genes, gene sets and surface markers shown in **d**; right, breakdown of participant phenotype (PIC or no PIC) and pathogen specificities (HIV, CMV) on UMAP plot as shown in **b**. *P* values reported above plots from χ² tests. **d**, Bubble plot comparing *Z*-scaled mean normalized expression and detection rates for curated surface markers, transcripts (italics) and gene signatures supporting cluster annotations, as detailed in Methods. **e**,**f**, Volcano plots summarizing differentially expressed genes (**e**) and surface proteins (**f**) among HIV-specific CD8+ T cells from controllers (blue) and non-controllers (red). **g**, Summary of top ten most significantly upregulated and downregulated gene set subnets from GSNA of HIV-specific CD8+ T cells from controllers versus non-controllers. Schematic in **a** was created using BioRender (https://biorender.com). HTO, hashtag oligonucleotide antibodies; FACS, fluorescence-activated cell sorting; 10X, single-cell encapsulation via 10X Genomics platform; NGS, next-generation sequencing; GEX, ADT, and VDJ represent gene expression, antibody-derived tags (CITE-seq) and TCR libraries, respectively.

indexing of transcriptomes and epitopes by sequencing (CITE-seq) analyses of 15,466 pHLA multimer-stained cells from controllers and non-controllers (Fig. 3a and Supplementary Data 1). Multi-modal

clustering of all samples based upon differential gene expression and surface markers revealed eight clusters, which were annotated manually on the basis of differentially expressed genes, gene sets and surface

markers (Fig. 3b and Extended Data Figs. 2 and 3a,b). Cluster 0 was elevated among HIV-specific cells in non-controllers, whereas cluster 1 was comparable between groups and clusters 2–7 were elevated among HIV-specific cells in controllers (Fig. 3c). Cluster 0, which was associated with lack of PIC, expressed canonical $T_{EM}$ and exhaustion ($T_{EX}$) markers including CD45RO, PD-1, TIGIT and *TOX* (Fig. 3d), indicating a potential role for T cell exhaustion in decreased functionality of HIV-specific CD8[+] T cells among non-controllers. By contrast, the PIC-associated cluster 6 expressed canonical $T_{SCM}$ genes and surface proteins associated with stemness, including CD45RA, CD62L, CCR7, CD27 and *TCF7* (Fig. 3d)[29], consistent with our flow cytometric analyses (Fig. 2h,i). This $T_{SCM}$-like cluster exhibited low inhibitory receptor expression, elevated oxidative phosphorylation gene signatures and increased surface expression of CD73 (Fig. 3d). T cells expressing CD73—an ectonucleotidase with previously reported roles in regulating metabolism through nicotinamide adenine dinucleotide modulation[30]—have been associated previously with spontaneous HIV control and reduced exhaustion[31,32]. PIC was also associated with $T_{EM}$-like cells expressing interferon response genes (cluster 5) and $T_{SCM}$-like cells co-expressing signatures of glycolysis that share features of transitory cells derived from stem-like precursors[33] (cluster 4; Fig. 3d–g, Extended Data Fig. 2 and Supplementary Data 2). Unlike $T_{EX}$ cells, which express effector-like signatures but are impaired for glycolysis, oxidative phosphorylation and proliferative potential[34], metabolic signatures elevated in T cells from controllers have been proposed previously to prime them for rapid signalling in response to antigen[35]. These data indicate HIV-specific CD8[+] T cells in controllers are characterized by molecular signatures of stemness, reduced exhaustion and metabolic fitness.

## Augmented stemness is associated with existing clones

We next investigated longitudinal changes following bNAb administration to define molecular signatures associated with the modest but significant augmentation of CD8[+] T cell stemness and proliferative capacity observed (Fig. 2c,i). As broadening of response specificities was not associated with PIC (Fig. 1c), we evaluated longitudinal changes within HIV epitope-specific responses targeted before intervention. PIC was not associated uniquely with diversification or expansion of T cell receptor (TCR) clonotypes following intervention (Fig. 4a and Supplementary Data 3). Epitope-specific responses were oligoclonal, with more than half of each response comprising one or two dominant clonotypes and without substantial emergence of new clonotypes following bNAb administration (Fig. 4b and Extended Data Fig. 3c,d).

By flow cytometry, we observed no significant increases in frequencies of HIV epitope-specific CD8[+] T cells (Fig. 4c), their activation measured by CD38 and HLA-DR co-expression (Fig. 4d), their in vivo proliferation marked by Ki-67 expression (Fig. 4e) or their cytotoxic differentiation measured by perforin and granzyme B co-expression (Fig. 4f). These results indicate a lack of peripheral response to antigen at the time points studied, which preceded waning of bNAb concentrations to subtherapeutic levels and detectable HIV recrudescence. By multi-modal single-cell analyses, we observed modest but significant upregulation of both CD45RA and CD62L surface marker expression following bNAb administration, consistent with increases in $T_{SCM}$ frequencies observed by flow cytometry (Fig. 2i), and an increase in gene signatures of oxidative metabolism (Fig. 4g–i and Supplementary Data 4), which has been associated previously with spontaneous control of HIV[36]. Following bNAb administration, we also observed small increases in the frequencies of $T_{SCM}$ and CD127[+]CD73[+] cell clusters, which have been associated previously with proliferative long-lived memory[37,38] and share gene signatures with follicular CD8[+] T cells in lymphoid tissues[39] (Fig. 4j,k). Although pre-existing differences in stemness better distinguished controllers from non-controllers than longitudinal changes (Figs. 2c, 3d and 4j), our results indicate that augmentation of CD8[+] T cell stemness in peripheral circulation following bNAb administration may involve CD8[+] T cell recirculation from lymphoid tissue sites of early bNAb-suppressed virus re-emergence, consistent with previous results in non-human primates[6].

## Discussion

In this study, we explored HIV-specific CD8[+] T cell responses in people with HIV on ART who received bNAbs and underwent concurrent or subsequent ATI. Examination of individuals who have remained mostly aviraemic without ART for up to 7 years from four similar interventional trials enabled us to investigate immune correlates of durable PIC at greater sensitivity than was feasible from individual trials. By evaluating cellular immunity at epitope-specific resolution using reagents matching autologous virus, our study also avoided potential confounding effects of immune escape. Our results indicate that HIV-specific CD8[+] T cells are more functional both before and after intervention in people who subsequently control viraemia without ART relative to those who receive the same intervention but experience viral rebound. HIV-specific CD8[+] T cells in post-intervention controllers were characterized by molecular and functional hallmarks of stemness, including the ability to proliferate, differentiate and mount cytotoxic recall responses against HIV antigens matched to autologous virus.

CD8[+] T cell stemness has been associated previously with spontaneous control of HIV viraemia, but its role in control of viraemia following treatment interruption is not well established. Class I HLA alleles associated with spontaneous HIV control do not seem to be associated with PTC[19,22,40]. Although HIV-specific CD8[+] T cell responses are dysfunctional in most people with HIV, and their functionality is not typically restored by ART[41], CD8[+] T cell functionality has been associated with case reports of PTC[42,43], and preservation of HIV-specific CD8[+] T cell functionality[44,45] may contribute to higher rates of PTC observed among early-treated people with HIV[19,20]. In addition, enhanced CD8[+] T cell functionality and stemness in some people following prolonged ART[46,47] may also contribute to PTC in people with HIV treated during chronic infection. However, as CD8[+] T cell responses to recrudescent viraemia typically lag HIV replication, they are probably insufficient to prevent rebound viraemia in most non-interventional ATI settings. Consistent with this, CD8[+] T cell responses are not typically associated with time-to-rebound but rather are associated with setpoint viral loads[48]. As ART is re-initiated upon viral rebound in most ATI trials, the impact of CD8[+] T cells on viral load setpoint is not typically measured and PTC in non-interventional studies has been associated more frequently with autologous neutralization and innate immunity[22,23,40].

As HIV frequently escapes from autologous neutralizing antibodies, passive infusion of exogenous bNAbs, especially in combination, has enabled prolonged suppression of viraemia[2,49,50]. CD8[+] T cells have been implicated in durable PIC among bNAb recipients due to a proposed vaccinal effect by which antigen–antibody complexes lead to the stimulation of cellular immunity[1–9]. While modest augmentation of CD8[+] T cell proliferative capacity following bNAb administration was observed consistently in our study, this effect was neither unique to PIC nor associated with new responses or TCR clonotypes against known HLA-optimal epitopes. Instead, our results implicate precise features of HIV-specific CD8[+] T cells before intervention that are further enhanced by bNAb administration and are associated with subsequent PIC, including their stemness, proliferative capacity, recall cytotoxicity and metabolic fitness. Indeed, these features have been associated previously with superior HIV-specific CD8[+] T cell functionality in spontaneous HIV controllers[11,13,16–18,36], from whom CD8[+] T cells and exogenous bNAbs can synergize to elicit in vitro HIV suppression[51]. We hypothesize that, by limiting the rate and magnitude of HIV recrudescence, bNAbs allow functional CD8[+] T cell responses a better chance to contain early virus rebound in lymphoid tissues, mediating PIC after bNAbs wane below therapeutic concentrations.

Despite including participants from four trials, our study remained limited by sample availability in several aspects, including scope and

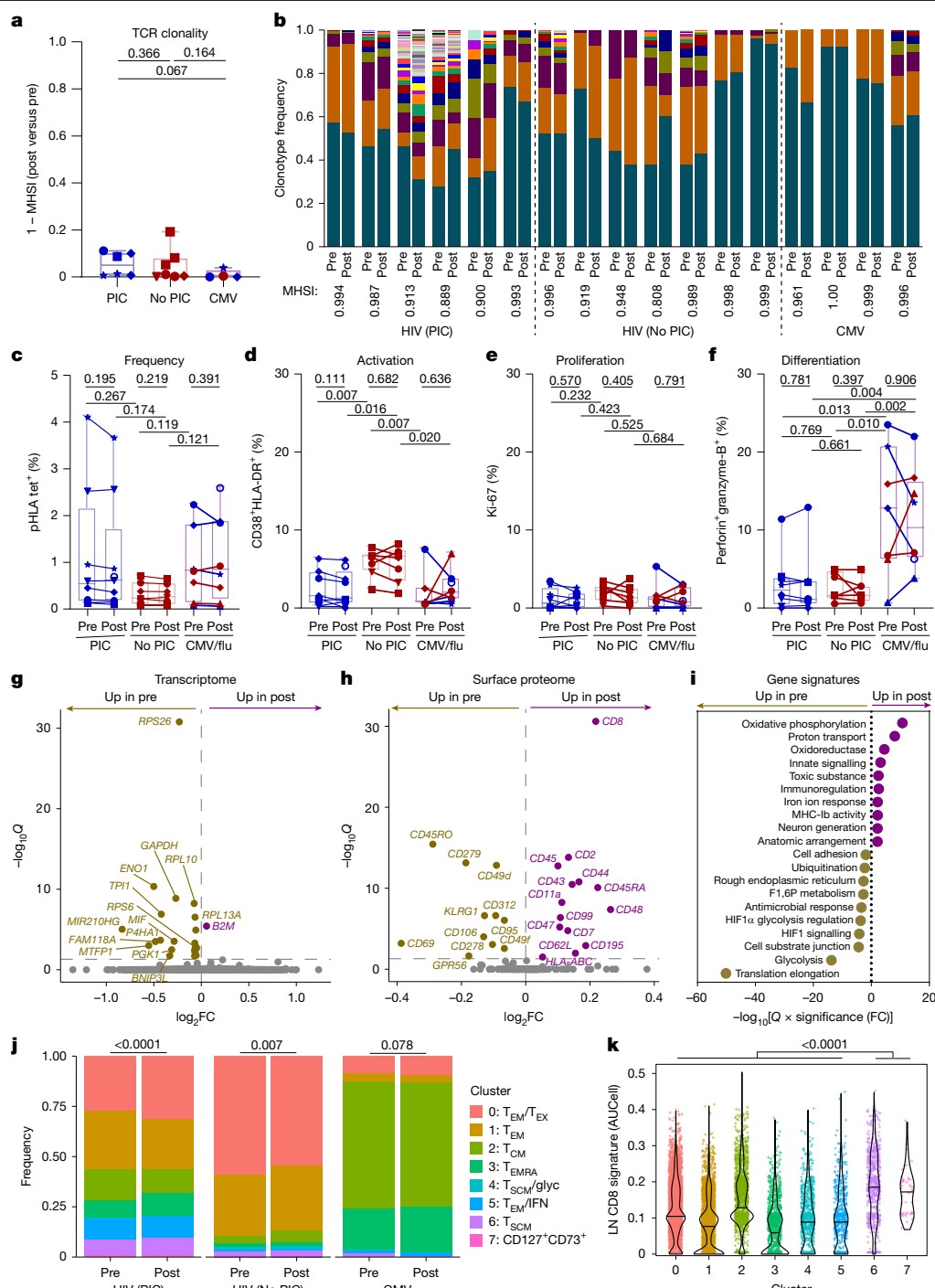

**Fig. 4 | Augmented CD8+ T cell stemness following bNAb administration is associated with pre-existing clonotypes. a**, Longitudinal TCR clonal diversification summarized as one minus Morisita-Horn similarity index (MHSI) among HIV-specific responses from controllers (blue, *n* = 6 responses) and non-controllers (red, *n* = 7 responses) or CMV-specific responses (*n* = 4). **b**, Longitudinal TCRβ CDR3 clonotypic frequencies and MHSI of HIV (*n* = 13) and CMV (*n* = 4) epitope-specific CD8+ T cell responses (paired columns) at pre- and post-bNAb time points from sorted pHLA tetramer+ cells, ordered and coloured by within-response rank for all responses with at least ten cells and longitudinal sampling and all clonotypes that occurred more than once in the data set; full data in Supplementary Data 3. **c**–**f**, Summaries of epitope-specific frequencies measured by pHLA tetramer (tet) staining among total CD8+ T cells (**c**), activation measured by surface CD38 and HLA-DR co-expression (**d**), proliferation measured by intranuclear Ki-67 (**e**) and cytotoxic effector differentiation measured by intracellular perforin and granzyme B co-expression (**f**) among

HIV pHLA tet+CD8+ T cell responses from controllers (*n* = 9 responses) and non-controllers (*n* = 7 responses), and among CMV/flu tet+CD8+ T cell responses (*n* = 8). **g**,**h**, Volcano plots summarizing longitudinal changes among HIV-specific CD8+ T cell responses from all participants with longitudinal sampling in gene (**g**) and surface protein (**h**) expression before (pre, gold) and after (post, magenta) intervention via CITE-seq analyses. **i**, Summary of top ten most significantly upregulated and downregulated gene set subnets from GSNA among HIV-specific CD8+ T cells from post- versus pre-intervention. **j**, Longitudinal cluster frequencies among HIV- and CMV-specific CD8+ T cells from controllers and non-controllers. **k**, Violin plot of single-cell AUCell expression levels of a gene signature associated with lymph node follicular CD8+ T cells[39] across clusters. Centre lines represent medians, boxes first and third quartiles, and whiskers ranges. Colour–symbol combinations represent participants (key in Table 1). *P* values reported above plots from two-sided Wilcoxon signed rank (**a** and **k**), two-sided paired (longitudinal) or unpaired (between-group) *t*-tests (**b**–**e**) and χ2 tests (**j**).

statistical power. As it was not feasible to screen CD8+ T cell responses using overlapping peptides spanning the entire HIV-1 proteome, we focused on known HLA-optimal epitopes matching autologous provirus sequence to facilitate downstream analyses using pHLA multimers. It is possible that our approach may have missed responses against as-yet undefined epitopes or those below our detection limit. Owing to limitations in specimen and pHLA multimer availability, we were able to profile only one-third of detected HIV-specific responses by cytometry and multiomics. As our study focused on HIV-specific CD8+ T cell responses, we did not evaluate other immune parameters that may contribute to PIC. Sampling of peripheral blood at a single post-intervention time point limited our ability to observe in vivo proliferative and cytotoxic responses to recrudescent viraemia. Owing to the retrospective and exploratory nature of our study, larger and prospective studies will be required to determine the predictive capacity of HIV-specific CD8+ T cell features preceding PIC. Studies investigating epitope-specific CD8+ T cell responses in lymphoid tissues, the primary sites of HIV persistence and recrudescence[52,53], and measurement of additional immune parameters such as autologous neutralization, innate immunity and HIV-specific CD4+ T cell responses, will be important to further delineate mechanisms of PIC.

Ongoing trials aim to elicit PIC in a larger proportion of people with HIV through improved or combinatorial interventions, including long-acting bNAbs[54], therapeutic vaccination[55] and agonists of cytokines such as IL-15 (ref. 56), which can rewire cellular metabolism of dysfunctional HIV-specific CD8+ T cells[57] and promote their migration to B cell follicles in lymphoid tissues[58]. Complementary new data emerging from two independent interventional trials further support a role for CD8+ T cell proliferation in PIC[55,59]. Our results suggest that immunotherapies capable of enhancing virus-specific CD8+ T cell stemness, proliferative capacity and recall cytotoxicity may greatly enhance durable HIV remission elicited by bNAb administration.

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

## Methods

### Study participants

We obtained approximately 40–80 million cryopreserved PBMCs from participants in the previously reported MCA-906 (NCT02825797), MCA-965 (NCT03526848), eCLEAR (NCT03041012) and TITAN (NCT03837756) trials[1-4], including seven controllers, who achieved PIC and maintained undetectable or very low plasma viral loads for more than 30 weeks (up to 7 years, and in some cases still ongoing) and five non-controllers, who experienced rebound viraemia following investigational infusion of bNAbs 3BNC117 and/or 10-1074 (Table 1). Longitudinal samples were included for 11 of 12 participants based on specimen availability at time points immediately preceding (pre) or 6–12 weeks following (post) bNAb administration in the context of ATI. eCLEAR participant 107, from whom we only included a post-intervention sample, was excluded from all pre-intervention and longitudinal analyses and its inclusion or exclusion did not affect our conclusions. To avoid potentially confounding effects of viraemia, samples were selected such that viraemia was undetectable in all participants at the time points sampled, with rebound viraemia in PINCs occurring several weeks after collection of the post-intervention samples evaluated. Secondary use of biological specimens was approved by the Mass General Brigham Human Research Committee following informed consent obtained during the primary studies in accordance with all applicable regulations and guidelines.

### Peptides

Peptides matching autologous, HLA class I-optimal HIV epitopes were synthesized to a purity of at least 80% at the Mass General Brigham Peptide Research Core using automated solid-phase Fmoc/tBu chemistry followed by HPLC and MALDI-MS analysis[60].

### Autologous HIV epitope-specific CD8+ T cell mapping

Cryopreserved PBMCs were thawed at 37 °C, recovered in RPMI medium (Sigma-Aldrich) supplemented with 10% fetal bovine serum (FBS, Sigma), 10 mM HEPES, 100 U ml⁻¹ penicillin, 100 μg ml⁻¹ streptomycin and 292 μg ml⁻¹ L-glutamine (Fisher Scientific; R10) overnight, resuspended at $1 \times 10^6$ cells ml⁻¹ in R10, and plated at 200 μl per well in Immobilon-P 96-well microtiter plates (Millipore) pre-coated with 2 μg ml⁻¹ anti-IFNγ (clone DK1, Mabtech). Individual HLA-optimal HIV-1 peptides matched to each subject's *HLA* genotype and autologous provirus sequence[27], where available, or for Clade B consensus sequence where unavailable (Supplementary Data 1), were added at 1 μM and incubated at 37 °C overnight. Triplicate negative control wells did not receive peptide and positive control wells were treated with 1 μg ml⁻¹ anti-CD3 (clone OKT3, Biolegend) and 1 μg ml⁻¹ anti-CD28 (clone CD28.8, Biolegend) antibodies. ELISpot assays were performed following manufacturer's protocol via biotinylated anti-IFNγ (clone B6-1, Mabtech) detection, streptavidin-ALP (Mabtech) and AP-conjugated substrate (Bio-Rad) followed by disinfection with 0.05% Tween-20 (Thermo Fisher) and analysis using CTL ImmunoSpot Analyzer Pro v.7.0.38.16. Responses greater than ten spots per well (50 spots per 10⁶ PBMCs) and threefold above negative controls were scored as positive.

### Proliferation

Cryopreserved PBMCs were thawed at 37 °C, recovered in R10 medium overnight, then stained at 37 °C for 20 min with 0.5 μM CellTrace CFSE (Thermo Fisher) as per manufacturer's protocol. Cells were then quenched and washed twice with R10 medium, resuspended at $1 \times 10^6$ cells ml⁻¹ in R10, and plated at 200 μl per well in 96-well round-bottom polystyrene plates (Corning). Individual HLA-optimal HIV-1 peptides matching each response detected previously by IFNγ ELISpot were added at 1 μM to triplicate wells and incubated at 37 °C for 6 days before flow cytometric assessment. Triplicate negative control wells did not receive peptide and positive control wells received 1 μg ml⁻¹ anti-CD3 (clone OKT3, Biolegend) and 1 μg ml⁻¹ anti-CD28 (clone CD28.8, Biolegend) antibodies. On day 6, cells were stained using Live/Dead Violet viability dye (Thermo Fisher, 10⁻³ dilution), AlexaFluor700-anti-CD3 (clone SK7, Biolegend, 10⁻² dilution) and APC-anti-CD8 (clone RPA-T8, Biolegend, 10⁻² dilution), then analysed by flow cytometry. Reported values for each epitope-specific response represent means of background-subtracted triplicates.

### Recall cytotoxicity

Recall cytotoxicity of HIV-1 epitope-specific memory CD8+ T cell responses was measured using the expanded antigen-specific elimination assay (EASEA) as per our published protocol[28]. In brief, PBMCs were rested overnight in R10 then incubated with 100 ng ml⁻¹ individual HLA-optimal HIV-1 peptide for 6 days to expand antigen-specific effector cells. Target CD4+ T cells were isolated from PBMC by negative magnetic separation (StemCell Technologies), activated in 24-well non-treated polystyrene plates (Corning) pre-coated with 2 mg ml⁻¹ anti-CD3 (clone OKT3, Biolegend) at 1–2 million cells ml⁻¹ in R10 with 2 mg ml⁻¹ anti-CD28 (clone CD28.2, Biolegend) and 50 U ml⁻¹ IL-2 (Peprotech) at 37 °C overnight, then expanded in tissue culture-treated 24-well plates (Corning) at 2 million cells ml⁻¹ in R10 with 50 U ml⁻¹ IL-2 at 37 °C for 5 days; 50% of target cells were pulsed for 30 min at 37 °C with 10 μM peptide and labelled with CellTrace Far Red dye (Thermo Fisher, 10⁻³ dilution) and mixed with unpulsed target cells 1:1, then labelled with CellTrace Violet dye (Thermo Fisher, 10⁻³ dilution). After 6 days of expansion, CFSE-labelled effector CD8+ T cells were isolated from pooled mononuclear cells by negative magnetic separation (StemCell Technologies) and co-cultured with target cells at effector to target (E:T) ratios of 0:1, 1:1, 2:1, 4:1 and 8:1 with 50,000 target cells per well in a treated 96-well polystyrene plate (Corning) for 4 h. Effector-only populations were stained with APC-conjugated pHLA tetramers (1:50 dilution) and all samples were stained with BV605-anti-CD3 (clone UCHT1, Biolegend, 10⁻² dilution), BUV395-anti-CD8 (clone RPA-T8, BD Biosciences, 10⁻² dilution), BV711-anti-CD4 (clone RPA-T4, Biolegend, 10⁻² dilution) and Live/Dead Near-IR (Thermo Fisher, 10⁻³ dilution) then analysed by flow cytometry. Results were gated as described previously, and percent elimination and AUC values were calculated as described previously[18,28].

### Phenotypic cytometry

Peptide-HLA monomers for immunodominant responses (listed in Supplementary Data 1) were purchased from ImmunAware as feasible. pHLA combinations were first validated for predicted binding using netMHCpan-4.0 (ref. 61) and successful complex folding was validated experimentally by the manufacturer at the time of production. Tetramers were produced by multimerization with APC-conjugated streptavidin (Biolegend) as per manufacturer's protocol. Staining was performed using 4 nM individual APC-conjugated pHLA tetramers at 4 °C for 30 min after 30-min pre-treatment with 50 nM dasatinib to prevent in vitro cell activation and activation-induced cell death. Cells were then stained with Live/Dead Near-IR viability dye (Thermo Fisher, 10⁻³ dilution), RB705-anti-CD3 (clone UCHT1, BD Biosciences, 10⁻² dilution), BV711-anti-CD8 (clone RPA-T8, Biolegend, 10⁻² dilution), BUV395-anti-CD45RA (clone HI100, BD Biosciences, 10⁻² dilution), RB780-anti-CD62L (clone DREG-56, BD Biosciences, 10⁻² dilution), PE-Dazzle 594-anti-CD38 (clone HB7, Biolegend, 10⁻² dilution) and BUV805-anti-HLA-DR (clone G46-6, BD Biosciences, 10⁻² dilution) for 30 min at 4 °C before fixation and permeabilization with eBiosciences Foxp3 transcription factor staining kit (Thermo Fisher) as per the manufacturer's protocol, followed by intracellular staining for PE-anti-perforin (clone B-D48, Biolegend, 1:50 dilution), FITC-anti-granzyme B (clone GB11, Biolegend, 1:50 dilution), and intranuclear staining for BV421-anti-Ki-67 (clone Ki-67, Biolegend, 1:50 dilution). Data were acquired using a FACSSymphony A5 cytometer and FACSDiva v.9.2 (BD) and analysed using FlowJo.

## Single-cell multiomics

Cryopreserved PBMCs were thawed and rested overnight before negative-selection magnetic CD8[+] T cell isolation (StemCell Technologies), pre-treated for 30 min with 50 nM dasatinib (Selleck Chemicals), then stained with 4 nM APC, PE, or BV421-conjugated pHLA tetramers [prepared using Total-Seq C barcode-conjugated streptavidin (Biolegend) and pHLA monomers described and validated above (Immunaware), listed in Supplementary Data 1], Total-Seq C Human Universal Cocktail v2.0 (Biolegend) as per manufacturer's protocol, BV711-anti-CD8 (clone RPA-T8, Biolegend, $10^{-2}$ dilution) and unique Total-Seq C hashing antibodies (Biolegend, 1:200 dilution). CD8[+] T cells from an HLA-mismatched individual were included for estimation of nonspecific barcoded tetramer binding and sorting gates were set above this level. Cells were washed using a HT2000 laminar cell washer (Curiox) then resuspended in 2% FBS in PBS with Sytox Green viability dye (Thermo Fisher). Viable pHLA[+] CD8[+] T cells were isolated by fluorescence-activated cell sorting (FACS, counts in Supplementary Data 1) into a single pool then encapsulated after splitting across four GEM-wells using Chromium GEM-X (10x Genomics) for CITE-seq. Gene expression (GEX), surface protein expression (antibody-derived tags, ADT), and TCR (VDJ) libraries were generated using the 10x Chromium GEM-X Single Cell 5′ v3 Dual Index kit with feature barcode technology (10x Genomics) following the manufacturer's protocol. Libraries were pooled at a 5:1:1 GEX to ADT to VDJ ratio and sequenced via paired-end reads on a NextSeq 2000 instrument with a 100-cycle P3 kit (Illumina).

Base-calling was performed using bcl2fastq and initial data-processing was performed using the Cell Ranger multi-analysis pipeline v.9.0.0 using refdata-gex-GRCh38-2020-A as a transcriptome reference and refdata-cellranger-vdj-GRCh38-alts-ensembl-5.0.0 as a VDJ reference. GEX, antibody capture (ADT) and TCR (VDJ) libraries were specified in the multi-analysis config file. Surface protein barcodes and hashtag barcodes corresponding to samples were designated as 'Antibody Capture' in the feature-reference file. After processing by Cell Ranger, the count matrix in sample_filtered_feature_bc_matrix was analysed using Seurat v.5.3.0 in R v.4.3.1. Hashtag and pHLA specificity-level sample demultiplexing was performed using the HTOdemux() function of Seurat, and cells were removed for which HTO_classification.global was not 'Singlet', hence removing cells with several or no hashtags. Cells for which pHLA barcodes were not detected were also removed, unless their corresponding TCR sequence matched expanded clones (more than five cells) from the data set, in which case they were reassigned to their matching specificity (1,757 total reassigned cells). A total of 25,866 HIV-specific and CMV-specific cells were recovered, of which 15,466 passed filtering (Supplementary Data 1). The GEX library yielded 239 mean variable unique genes per cell, and 751 mean unique molecular identifiers per cell. The ADT library yielded 522 mean unique molecular identifiers per cell. To avoid clustering driven by clonotype-specific TCR gene expression, gene features for which the symbols matched the regular expression '^TR[ABDG][VJC]' were removed from the data set before clustering[62]. Using the Seurat function FindVariableFeatures(), 4,000 variable genes were selected for dimensionality reduction and differential expression analysis. Counts were log normalized, scaled and centred before dimensionality reduction and clustering. Clustering was performed using weighted nearest-neighbours (WNN) clustering via Seurat's FindNeighbors() and FindClusters() functions with the argument resolution = 0.35.

Differential expression was performed using Seurat's FindMarkers() function using default parameters, including Wilcoxon tests for statistical significance. Pathway analysis was performed using the tmodCERNOtest() from the tmod R package v.0.46.2 (ref. 63) using a subset of MSigDB version v.7.5.1 (ref. 64) that included hallmark, gene ontology, reactome, KEGG, biocarta and wikipathways gene sets. Primary cluster annotations as $T_{EM}$, central memory T ($T_{CM}$) cells, $T_{SCM}$ and terminally differentiated memory ($T_{EMRA}$) were defined using CD45RA/RO and CD62L expression for comparability to flow cytometry results. Primary and secondary cluster annotations were also supported by differentially expressed surface ADTs corresponding to CCR7, CD127, CD226, PD-1, TIGIT, CX3CR1 and CD73; differentially expressed transcripts corresponding to *TCF7*, *TOX*, *GZMB*, *GZMK*, *GAPDH*, *ENO1* and *IFITM1*; and differentially expressed gene sets corresponding to aerobic glycolysis (WP4628), oxidative phosphorylation (M12919), interferon alpha response (M5911), lymph node follicular CD8[+] T cells (*CXCR5*, *SLAMF6*, *SELL*, *TCF7*, *ID3*, *CD200*, *ICOS*, *IL7R* and *BCL6*)[39] and T cell activation (M2810), which were quantified via AUCell[65] and plotted as bubble and/or violin plots in R. Gene set network analysis was performed using the GSNA R package, v.0.1.4.9, as described previously[15,18]. Longitudinal differential expression analyses were performed across HIV-specific responses from all participants with longitudinal sampling.

TCR clonotypes were assigned based on *TRB* CDR3 sequences and those appearing only once in the data set were excluded from clonotypic analysis. Diversity of clonotypes within a sample was quantified using Simpson diversity index and similarity of clonotypic composition between longitudinal samples was quantified using MHSI[66], whereas its inverse (1 − MHSI) was used to assess longitudinal clonotypic divergence. MHSI measures overlap of clonotype proportions between two samples on a scale from 0 (no similarity) to 1 (identical) and is relatively robust to differences in sample size. Extended analyses are reported in Supplementary Data 3.

## Statistical analyses, reproducibility and figure preparation

Analyses in this exploratory study were primarily descriptive and hypothesis-generating. Statistical analyses were performed using GraphPad Prism v.10.4 and R. Normality was estimated using Shapiro–Wilk tests. Normally distributed data were compared using *t*-tests and non-normally distributed data were compared using Wilcoxon signed rank tests and Spearman correlations. All replicate measurements reflect distinct biological samples or epitope-specific responses, as specified in each figure legend. All representative data shown are accompanied by summary data encompassing the entire data set, with the precise number of biological replicates specified in each figure legend. All statistical tests were two-tailed. Wherever box-and-whisker diagrams are depicted, centre lines represent medians, ticks represent means, boxes represent first and third quartiles, and whiskers represent ranges. Figures were prepared using Adobe Illustrator v.29.8.2, GraphPad Prism, R and BioRender (https://biorender.com).

## Reporting summary

Further information on research design is available in the Nature Portfolio Reporting Summary linked to this article.

## Data availability

Full single-cell multiomics data are available from the NCBI Gene Expression Omnibus (GEO: GSE294440). The GRCh38 reference genome is available from NCBI GenBank (GCA_000001405.15). MSigDB gene set references can be obtained from https://data.broadinstitute.org/gsea-msigdb/msigdb/release/7.5.1/. The remaining data are included within the Article and Supplementary Data 1–4.

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

**Acknowledgements** We are grateful to the study participants; to eCLEAR, TITAN, MCA-906 and MCA-965 study clinicians and staff; and to J. Hitschfel, E. Çakan, J. Chen, R. Olsen and G. Schmidt Frattari for helpful discussions and advice. This work was supported by funding from United States National Institutes of Health (AI184606, AI155233, AI152979, AI176579, AI44462), Howard Hughes Medical Institute and the Lundbeck Foundation (R381–2021–1405). Funders had no role in study design, data collection and analysis, decision to publish or preparation of the manuscript.

**Author contributions** D.R.C. designed the study with input from B.D.W., M.C.N., O.S.S., M.C. and J.D.G. M.C., J.D.G., O.S.S. and M.C.N. provided specimens. Z.K., H.W., M.J.O., D.Y.C. and D.R.C. performed experiments supported by technical contributions from J.A.A., A.P.-T. and N.B. and critical reagents from A.K. J.M.U. and D.R.C. analysed data. M.L. provided autologous provirus sequences. D.R.C. and B.D.W. supervised the work and obtained funding. D.R.C. wrote the initial draft. All authors contributed to the final draft.

**Competing interests** The authors declare no competing interests.

**Additional information**
**Correspondence and requests for materials** should be addressed to David R. Collins.

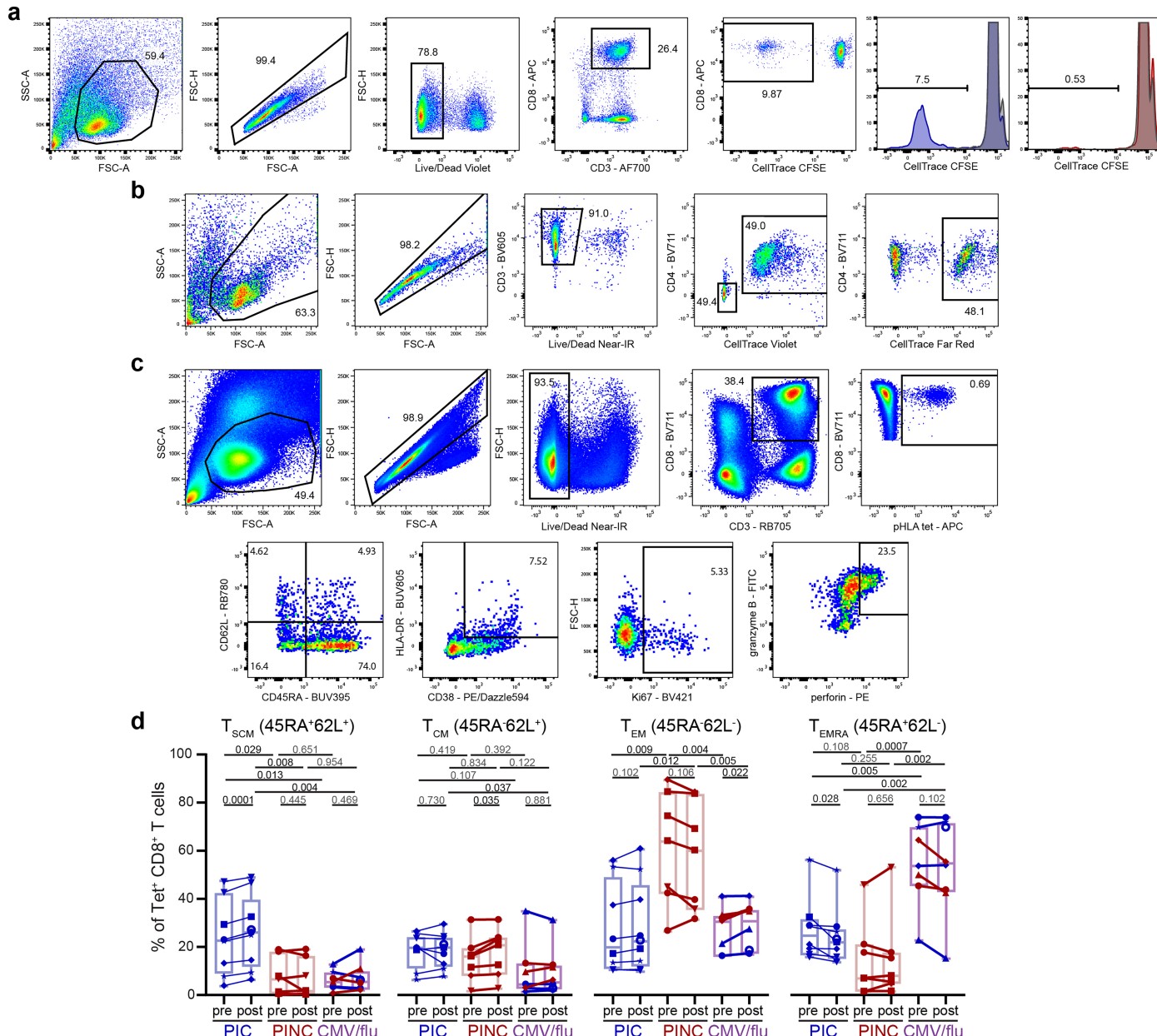

**Extended Data Fig. 1 | Flow cytometric CD8⁺ T cell profiling. (a-c)** Representative gating schema for measurement of epitope-specific proliferation (a), elimination of peptide-pulsed (CellTrace Far Red⁺) CD4⁺ T cell targets by peptide-expanded CD8⁺ T cell effectors (b), and phenotypic profiling of pHLA tetramer⁺ (Tet⁺) cells (c) by flow cytometry. Panel a also includes representative proliferation histogram overlays for HIV epitope-specific responses from PIC 5120 (blue) and PINC 9243 (red) relative to unstimulated controls (gray). (**d**) Memory subset frequencies among HIV Tet⁺ CD8⁺ T cell responses from controllers (PIC, $n$ = 9 responses) and non-controllers (PINC, $n$ = 7 responses), and among CMV/flu Tet⁺ CD8⁺ T cell responses from both groups ($n$ = 8). Center lines represent medians, ticks represent means, boxes represent first and third quartiles, and whiskers represent ranges. Color-symbol combinations represent participants (key in Table 1). $P$-values reported above plots from two-sided paired (longitudinal) or unpaired (between-group) $t$-tests.

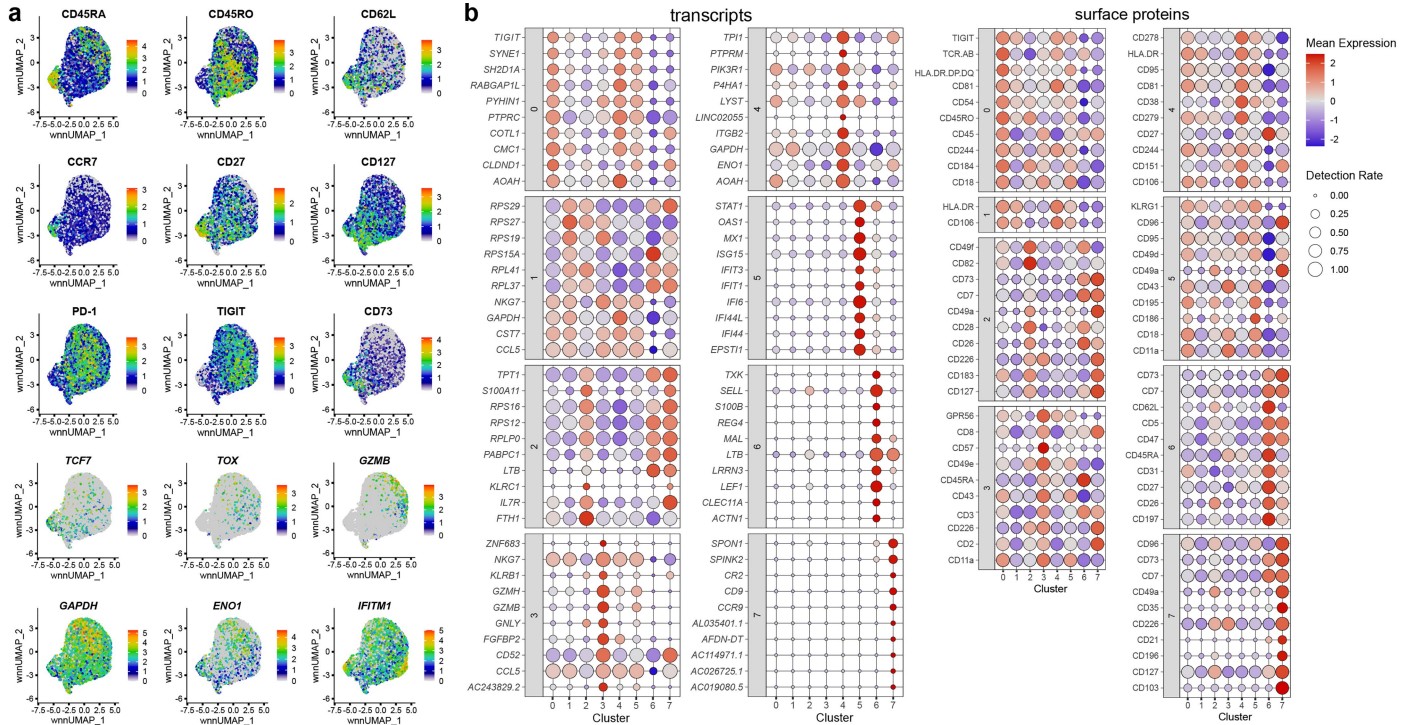

**Extended Data Fig. 2 | Differential expression between clusters. (a)** Feature plots of expression levels of selected differentially expressed surface proteins and transcripts (italics) projected onto UMAP plots, supporting cluster annotations in Fig. 3. **(b)** Bubble plots of z-scaled mean normalized expression and detection rates for top differentially expressed transcripts (left) and surface proteins (right) upregulated in each cluster, ranked by adjusted *p* value.

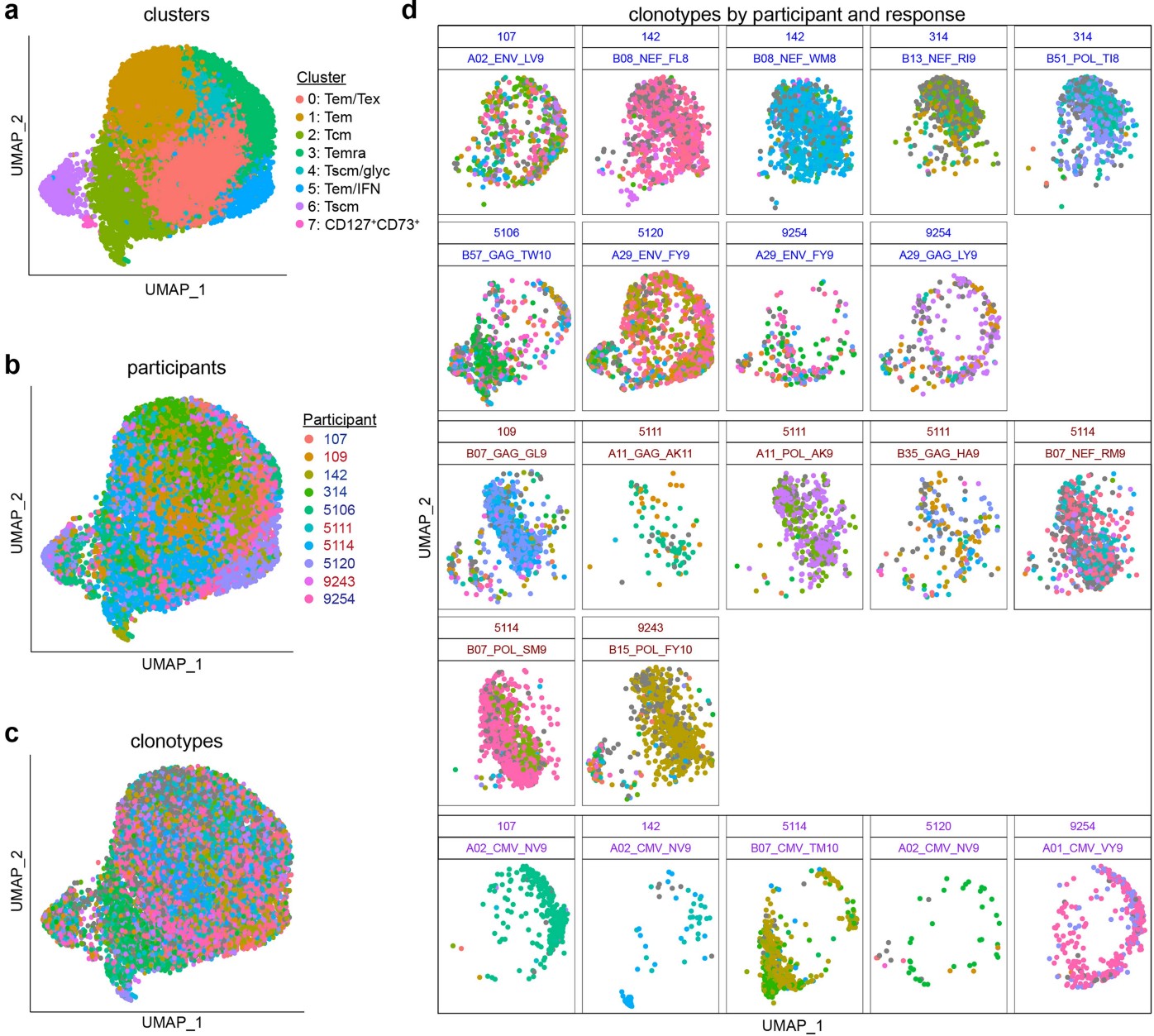

**Extended Data Fig. 3 | Multimodal clustering and TCR clonotypes.**
(**a-d**) UMAP of HIV and CMV epitope-specific CD8⁺ T cells colored by WNN cluster (a), participant (b), *TRB* CDR3 clonotype (c), or *TRB* CDR3 clonotype separated by participant and response (d). Gray points represent singlets, whereas colored points are clonally expanded.

# Reporting Summary

## Statistics

For all statistical analyses, confirm that the following items are present in the figure legend, table legend, main text, or Methods section.

| n/a | Confirmed | |
|---|---|---|
| ☐ | ☒ | The exact sample size (*n*) for each experimental group/condition, given as a discrete number and unit of measurement |
| ☐ | ☒ | A statement on whether measurements were taken from distinct samples or whether the same sample was measured repeatedly |
| ☐ | ☒ | The statistical test(s) used AND whether they are one- or two-sided *Only common tests should be described solely by name; describe more complex techniques in the Methods section.* |
| ☒ | ☐ | A description of all covariates tested |
| ☐ | ☒ | A description of any assumptions or corrections, such as tests of normality and adjustment for multiple comparisons |
| ☐ | ☒ | A full description of the statistical parameters including central tendency (e.g. means) or other basic estimates (e.g. regression coefficient) AND variation (e.g. standard deviation) or associated estimates of uncertainty (e.g. confidence intervals) |
| ☐ | ☒ | For null hypothesis testing, the test statistic (e.g. *F*, *t*, *r*) with confidence intervals, effect sizes, degrees of freedom and *P* value noted *Give P values as exact values whenever suitable.* |
| ☒ | ☐ | For Bayesian analysis, information on the choice of priors and Markov chain Monte Carlo settings |
| ☐ | ☒ | For hierarchical and complex designs, identification of the appropriate level for tests and full reporting of outcomes |
| ☐ | ☒ | Estimates of effect sizes (e.g. Cohen's *d*, Pearson's *r*), indicating how they were calculated |

*Our web collection on statistics for biologists contains articles on many of the points above.*

## Software and code

Policy information about availability of computer code

| Data collection | IFNG elispot data collection was performed using CTL ImmunoSpot Analyzer Pro version 7.0.38.16. Flow cytometric data collection and FACS were performed using BD FACSDiva version 9.2. |
|---|---|
| Data analysis | Flow cytometric data analyses were performed using FlowJo version 10.10.0. Statistical analyses were performed using GraphPad Prism version 10.4 and R version 4.3.1. Single-cell multiomics data analyses were performed using R version 4.3.1, cellranger version 9.0.0, bcl2fastq version 2.20, seurat version 5.3.0, tmod version 0.46.2, and GSNA version 0.1.4.9. Data visualizations were prepared using ggplot2 version 3.5.2 and Adobe Illustrator version 29.8.2. |

For manuscripts utilizing custom algorithms or software that are central to the research but not yet described in published literature, software must be made available to editors and reviewers. We strongly encourage code deposition in a community repository (e.g. GitHub). See the Nature Portfolio guidelines for submitting code & software for further information.

## Data

Policy information about availability of data

All manuscripts must include a data availability statement. This statement should provide the following information, where applicable:
- Accession codes, unique identifiers, or web links for publicly available datasets
- A description of any restrictions on data availability
- For clinical datasets or third party data, please ensure that the statement adheres to our policy

Full single-cell multiomics data are available via the NCBI Gene Expression Omnibus (GEO) via accession number GSE294440. The GRCh38 reference genome is available at NCBI GenBank via accession number GCA_000001405.15. MSigDB gene set references can be obtained from https://data.broadinstitute.org/gsea-msigdb/msigdb/release/7.5.1/. The remaining data are included within the manuscript and supplemental materials.

## Research involving human participants, their data, or biological material

Policy information about studies with human participants or human data. See also policy information about sex, gender (identity/presentation), and sexual orientation and race, ethnicity and racism.

| | |
|---|---|
| Reporting on sex and gender | Biological sex of each participant is reported in Table 1 as previously published for each parent trial. |
| Reporting on race, ethnicity, or other socially relevant groupings | Race (American Indian, Black, or White) and ethnicity (Hispanic or not Hispanic) of each participant are reported in Table 1 as previously published for each parent trial. |
| Population characteristics | Age, class-I HLA genotypes, and clinical histories related to HIV are reported for each participant in Table 1 as previously published for each parent trial. |
| Recruitment | This study includes only secondary use of previously collected samples. |
| Ethics oversight | Secondary use protocols were approved by the Mass General Brigham Human Research Committee |

Note that full information on the approval of the study protocol must also be provided in the manuscript.

# Field-specific reporting

Please select the one below that is the best fit for your research. If you are not sure, read the appropriate sections before making your selection.

☒ Life sciences ☐ Behavioural & social sciences ☐ Ecological, evolutionary & environmental sciences

For a reference copy of the document with all sections, see nature.com/documents/nr-reporting-summary-flat.pdf

# Life sciences study design

All studies must disclose on these points even when the disclosure is negative.

| | |
|---|---|
| Sample size | Sample sizes were constrained by specimen and reagent availability. |
| Data exclusions | Only participants who received intervention were included in the analyses. Participants/responses for which only one longitudinal sample was measured were excluded from longitudinal statistical comparisons. Doublets were excluded from multimodal single-cell analyses based on hashing and tetramer oligonucleotides. Cells with TCRs that occurred only once and cells for which TCR sequences were not detected were excluded from TCR clonotypic analyses. |
| Replication | Proliferation assays were confirmed in triplicate and averaged. Metrics were also repeated across longitudinal samples for each participant. The precise number of biological replicates is specified for each experiment in the figure legends and each data point is displayed in the figures. Representative data are only shown adjacent to the corresponding full data set for illustrative purposes. Further replication beyond those listed here were prohibited by limited specimen availability. |
| Randomization | This manuscript reports secondary analyses of specimens from previous trials. Experimental groups (PIC, PINC) were determined based on the presence or absence of prolonged virologic control without resumption of ART, as previously reported by each parent trial. Longitidinal samples (pre, post) were pre-determined based on which samples were collected prior to or following intervention in the parent trials. Viremia as a potential covariate was controlled by inclusion only of samples without detectable HIV viremia. Demographics are summarized in Table 1. Due to limited participant numbers, covariate modeling or controlling for additional potential covariates was not feasible. |
| Blinding | As this manuscript reports secondary analyses of specimens from previous trials, formal blinding was not part of the study design. |

# Reporting for specific materials, systems and methods

We require information from authors about some types of materials, experimental systems and methods used in many studies. Here, indicate whether each material, system or method listed is relevant to your study. If you are not sure if a list item applies to your research, read the appropriate section before selecting a response.

## Materials & experimental systems

| n/a | Involved in the study |
|-----|------------------------|
| ☐ | ☒ Antibodies |
| ☒ | ☐ Eukaryotic cell lines |
| ☒ | ☐ Palaeontology and archaeology |
| ☒ | ☐ Animals and other organisms |
| ☒ | ☐ Clinical data |
| ☒ | ☐ Dual use research of concern |
| ☒ | ☐ Plants |

## Methods

| n/a | Involved in the study |
|-----|------------------------|
| ☒ | ☐ ChIP-seq |
| ☐ | ☒ Flow cytometry |
| ☒ | ☐ MRI-based neuroimaging |

## Antibodies

| | |
|---|---|
| Antibodies used | anti-IFN-g, clone DK1, Mabtech, cat# 3420-2A, lot# 161; anti-CD3, clone OKT3, Biolegend, cat# 317326, lot# B407799; anti-CD28, clone CD28.2, Biolegend, cat# 302934, lot# B374639; anti-IFN-g, clone B6-1, Mabtech, cat# 3420-2A, lot#161; AlexaFluor700-anti-CD3, clone SK7, Biolegend, cat# 344822, lot# B420037; APC-anti-CD8, clone RPA-T8, Biolegend, cat# 301014, lot# B386144; BV605-anti-CD3, clone UCHT1, Biolegend, cat# 300460, lot# B430690; BUV395-anti-CD8, clone RPA-T8, BD Biosciences, cat# 563795, lot# 4292914; BV711-anti-CD4, clone RPA-T4, Biolegend, cat# 300558, lot# B420968; RB705-anti-CD3, clone UCHT1, BD Biosciences, cat# 570237, lot# 3229245; BV711-anti-CD8, clone RPA-T8, Biolegend, cat# 301044, lot# B425053; BUV395-anti-CD45RA, clone HI100, BD Biosciences, cat# 740298, lot# 5091519; RB780-anti-CD62L, clone DREG-56, BD Biosciences, cat# 569211, lot# 4200635; PE-Dazzle594-anti-CD38, clone HB-7, Biolegend, cat# 356630, lot# B406413; BUV805-anti-HLA-DR, clone G46-6, BD Biosciences, cat# 568335, lot# 4178322; PE-anti-perforin, clone B-D48, Biolegend, cat# 353304, lot# B397495; FITC-anti-granzyme B, clone GB11, Biolegend, cat# 515403, lot# B397296; BV421-anti-Ki-67, clone Ki-67, Biolegend, cat# 350506, lot# B356738; BV711-anti-CD8, clone RPA-T8, Biolegend, cat# 301044, lot# B425053; Total-Seq C Human Universal Cocktail v2.0, Biolegend, cat# 399910, lot# B408342; Total-Seq C anti-human hashtags 1-18, clone LNH-94/2M2, Biolegend, cat# 394661-394693, lot# B344497 |
| Validation | Species reactivity and suitability for each application were validated by the commercial suppliers (Biolegend, BD Biosciences, Mabtech) for each antibody, with quality control certification provided for each lot. |

## Plants

| | |
|---|---|
| Seed stocks | *Report on the source of all seed stocks or other plant material used. If applicable, state the seed stock centre and catalogue number. If plant specimens were collected from the field, describe the collection location, date and sampling procedures.* |
| Novel plant genotypes | *Describe the methods by which all novel plant genotypes were produced. This includes those generated by transgenic approaches, gene editing, chemical/radiation-based mutagenesis and hybridization. For transgenic lines, describe the transformation method, the number of independent lines analyzed and the generation upon which experiments were performed. For gene-edited lines, describe the editor used, the endogenous sequence targeted for editing, the targeting guide RNA sequence (if applicable) and how the editor was applied.* |
| Authentication | *Describe any authentication procedures for each seed stock used or novel genotype generated. Describe any experiments used to assess the effect of a mutation and, where applicable, how potential secondary effects (e.g. second site T-DNA insertions, mosiacism, off-target gene editing) were examined.* |

## Flow Cytometry

### Plots

Confirm that:

☒ The axis labels state the marker and fluorochrome used (e.g. CD4-FITC).

☒ The axis scales are clearly visible. Include numbers along axes only for bottom left plot of group (a 'group' is an analysis of identical markers).

☒ All plots are contour plots with outliers or pseudocolor plots.

☒ A numerical value for number of cells or percentage (with statistics) is provided.

### Methodology

| | |
|---|---|
| Sample preparation | Cryopreserved PBMCs were thawed at 37 C and rested overnight in RPMI + 10% FBS prior to each assay. |
| Instrument | Data were collected using BD FACSSymphony A5, LSR-II, and FACSAria instruments. |
| Software | Collection was performed using BD FACSDiva. Analysis was performed using FlowJo. |

| Cell population abundance | Abundances of each cell population/subpopulation are reported for all flow cytometry and multiomics data as frequencies in the figures, extended figures, and supplementary data. |
| Gating strategy | Intact live CD8+ cells were gated on forward and side scatter, viability dye, and CD8. Elimination assay data were pre-gated on intact, live, CTV+ target cells. Gates are represented in manuscript figures. |

☒ Tick this box to confirm that a figure exemplifying the gating strategy is provided in the Supplementary Information.

