## [Peer Review File · Nature]

CD8+ T cell stemness precedes post-intervention control of HIV viremia

Corresponding Author: Dr David Collins

Version 0:

Reviewer comments:

Referee #1

(Remarks to the Author)

Kiani and colleagues compared T cell responses in patients with or without durable post-intervention control following administration of broadly neutralizing anti-HIV-1 antibodies. They observed a stronger T cell response in patients with durable control, in terms of magnitude but not breadth (as defined by the number of recruited clones). Further characterization revealed that these cells were already present in higher numbers prior to the intervention, showed a higher frequency of proliferation-competent progenitor or stem-like cells, and demonstrated superior effector capacity in ex vivo assays. Overall, the manuscript presents these findings in a robust and convincing manner.

What limits my enthusiasm in supporting the manuscript for publication in Nature is the absence of major conceptual novelty or a transformative breakthrough. While it is certainly valuable to report that such T cell populations can be identified and may be predictive of durable control, the findings are largely in line with what one might expect given the current understanding in the field. As such, I find it difficult to identify data within the manuscript that would, in my view, meet the threshold for Nature.

Overall, the reported conclusions are well supported, and the study is well executed in a challenging area of research.

My only remaining major concern is that the authors suggest the identified features could be used to stratify patients; however, they have not demonstrated the predictive power of these features in a larger, independent cohort. One reason for raising this point is that the observed differences in response magnitude, while statistically significant, appear relatively modest. This raises questions about whether they are sufficient for reliable prediction of clinical outcomes. Demonstrating such predictive capacity would substantially enhance the impact and translational relevance of the study.

Referee #2

(Remarks to the Author)

General comments:

In this paper, the authors analyzed participants' samples and clinical data from four clinical trials involving infusion of the broadly neutralizing antibodies (bNAbs) 3BNC117 and 10-1074, followed by analytical treatment interruption (ATI). The study compared seven post-intervention controllers (PICs) and five post-intervention non-controllers (PINCs). Blood samples from pre- and post-intervention periods (6 or 12 weeks after ATI initiation, while viremia remained suppressed) were compared, focusing on HIV-specific CD8+ T cells. The authors hypothesize that by limiting the rate and magnitude of HIV recrudescence, bNAbs provide functional CD8+ T cell responses with a better opportunity to contain early viral rebound in lymphoid tissues. This mechanism may mediate PIC status after bNAb concentrations decline below therapeutic levels. They associate PIC with superior pre-intervention HIV-specific CD8+ T cell proliferative capacity, observed upon stimulation with autologous HIV peptides ex vivo, increased recall cytotoxicity against autologous HIV peptide-pulsed CD4+ T cells, enhanced HIV tetramer-positive CD8+ T cell stemness, improved metabolic fitness, and reduced T cell exhaustion. From

their data, the authors also conclude that there is no emergence of new specificities or clonotypes of HIV-specific CD8+ T cells.

The study is relevant to the field, presents novel and interesting findings, and represents a considerable effort to gather data from several interventional trials. However, several issues limit the significance of the study and require further examination. There are also some overstatements and conclusions that are not sufficiently supported by the data. Addressing the specific comments below could significantly improve the manuscript.

Specific major comments:

1. The experimental approach used to examine the patterns of epitopes targeted by HIV-specific CD8+ T cells pre- and post-bNAb administration is too narrow to support the sweeping conclusion that there is no emergence of new specificities or clonotypes. A notable strength of the study design is the use of autologous virus sequences. A weakness, however, is the use of limited panels of optimal epitopes matched to the HLA haplotypes of the participants. There are potential selection biases in such panels, which tend to preferentially include immunodominant epitopes elicited during infection in the absence of immunotherapy. Should bNAb administration lead to the emergence of new, less frequent specificities, they would be missed by this approach. The breadth of responses observed here—despite the use of autologous sequences (median of 3.5 epitopes)—suggests that detection is far from comprehensive, especially given that published studies using pools of overlapping peptides spanning the entire HIV proteome typically report a breadth of 10–30 CD8 epitopes, with around 15 specificities being common in ART-suppressed participants. Other studies on bNAb+ATI interventions suggest increased CD8+ T cell responses in both breadth (e.g., by ELISpot, as in this study) and magnitude (e.g., by ICS). Given how critical these conclusions are for understanding possible mechanisms of control, the authors should explore this issue further. While PBMC numbers may be limiting in some participants, the authors should confirm their findings in at least a subset of participants using a more inclusive technique.
2. Some important data and text are contradictory: While Figure 3c and Supplementary Fig. 2e support the claim that there is more pre-intervention stemness in PICs compared to PINCs, cluster 6 is one of the smallest and doesn't seem to reflect the flow data in Fig. 2h. Figure 3g shows upregulation of glycolysis pathways in PICs and enriched T cell cytotoxicity in PINCs, which contradicts the text. Supplementary Fig. 2e shows an increased frequency of exhausted T cells and Temra, not Tscm, in PICs post-intervention, again contradicting the text. Figure 4e contradicts the idea that PIC status is mediated by superior proliferation after bNAb concentrations decline below therapeutic levels.
3. Cluster annotation (Fig. 3, Extended Data Fig. 2) needs more supporting data: The study does not provide details on the gene lists and signatures used to annotate the clusters. The rationale for selecting the canonical markers for annotation is unclear. The authors should show the gene expression of canonical markers on the UMAP to better justify the cluster annotations. Clusters 0, 1, and 2 appear very similar on the heatmap, and the annotation requires further explanation. GZMB and CX3CR1 have been described in the context of T cell exhaustion (Tex); the authors' reasoning for annotating this cluster as TEMRA is not sufficiently explained. Similarly, for cluster 4, TCF7 is not expressed while TOX is highly expressed, which goes against the literature on Tpex. Plotting the top 10 DEGs for each cluster (can be placed in the supplemental) would help. Once clusters are annotated with a phenotype, this should be used in the text, as readers do not remember cluster numbers.
4. The authors analyzed TCR clonality using the Morisita-Horn index to compare the similarity of pre- and post-intervention TCR clones (Figure 4). They conclude that there are no new epitopes in PICs post-intervention. While the high Morisita-Horn index values indeed do not suggest a major reshaping of the CD8+ T cell repertoire post-treatment, this should be more rigorously assessed. By visual inspection alone, it is not possible to determine the number of clones, and this should be formally tested. The Simpson diversity index could be used to measure the diversity within a single timepoint. Moreover, it is well established in the field that examining only the beta chain does not accurately describe a TCR clone. The authors should also use the alpha chain to perform clonal analysis. Using both CDR3 α and CDR3 β sequences to define a TCR clone, as opposed to using only CDR3 β , reduces the likelihood of falsely grouping TCR clones that happen to share the same beta chain. This approach captures a higher level of TCR diversity, as using only beta chains can lead to an overestimation of clonal expansion when distinct clones share the same beta chain.
5. More generally, and in line with comment #1: Given the small study groups (which is an expected limitation), the authors should be more nuanced when interpreting the lack of statistically significant differences between PIC and PINC for some of the parameters measured as proof of similarity or absence of effect of the bNAb+ ATI intervention.

Specific minor comments:

Figure 1:

- In addition to Table 1, the authors should provide a comprehensive table listing the sequence and number of successfully tested peptides for each donor.
- They should also indicate in Figures 1 and 2 which peptide elicited which response, and in which donor.
- Please check the figure legend for Fig. 1c—the numbers do not seem to add up.

Figure 2:

- Please add CFSE histogram overlays of undivided and proliferative peaks for representative PIC and PINC samples in the supplementary material.
- Please indicate the IDs of the two PINC donors with high recall cytotoxicity but low proliferation, and comment on why

these donors may behave differently.

- The authors state that recall cytotoxicity was strongly associated with proliferative capacity, supporting a role for highly functional HIV-specific CD8+ T cells in PICs. Was proliferative capacity also correlated with the IFN γ response?
- Given that PINCs hardly expand in response to peptide stimulation, it is surprising that the 5-day expansion yielded enough cells to perform the killing assays.

Use of tetramers and dextramers:

- Please provide a comprehensive list of the specific tetramers used for each donor, and explain the validation process.
- Were the same pHLA monomers used to generate the tetramers and dextramers?
- Please provide additional evidence to distinguish the Tscm population in Fig. 2g from naïve T cells (T_n).
- The data show that PICs have a higher frequency of HIV- and CMV-tetramer+ cells compared to PINCs at both timepoints (Fig. 4c). The authors should comment on the potential significance of this observation.
- The perforin staining shown in Supplementary Fig. 1c is not convincing—the “negative” population already appears quite bright.

scRNA-seq and CITE-seq on sorted HIV-specific CD8+ T cells:

In addition to the major comments already raised, please also address the following:

- Provide the staining profiles of the dextramer-sorted cells to validate the sorting strategy.
- Explain how the sample matrix was structured and how batch effects were corrected for in the scRNA-seq and CITE-seq analyses.
- Clarify the discrepancy in the number of PIC and PINC cells analyzed. Provide the number of cells per donor and per dextramer that were sequenced.
- Explain the metric used to integrate pathway analysis into the heatmap (e.g., z-score).
- Describe the statistical approaches used for contrast and pathway analyses, and discuss why the single-cell data resolution was not better integrated into these analyses.
- Clarify the observation that the CMV response is mainly driven by a TCM-like cluster, which is not apparent in the flow cytometry data in Extended Fig. 1d.
- For Fig. 4b, explain the rationale for showing only the CDR3b sequences of 6 out of 9 PIC dextramers and 4 out of 5 CMV dextramers. Clarify the number of epitope responses observed in each group (PIC, PINC, CMV), and reconcile any discrepancies with the information provided in Extended Fig. 2d.
- Can the authors clarify what Tpm refers to?

Referee #3

(Remarks to the Author)

I co-reviewed this manuscript with one of the reviewers who provided the listed reports.

Referee #4

(Remarks to the Author)

A. In this interesting manuscript, Kiani et al analyze the role of HIV-specific CD8+ T cell responses in post-intervention control. They compare the responses in 7 post-intervention controllers (PICs) to those of 5 post-intervention non-controllers (PINCs). They measure breadth of responses to autologous epitopes, proliferation, stemness, TCR clonotypes and differential expression of genes and surface proteins using single cell RNA seq. They show convincingly that CD8+ T cell stemness correlates with the PIC outcome.

B. CD8+ T cell stemness has been associated with elite control, but PICs are a different subset of patients who do not have the same level of protective HLA alleles. The study is also important because it provides mechanistic insight into a phenomenon that was reported in the SIV model, but not often seen in human studies. Thus, this is an original and highly significant finding.

C. The methodology is quite robust; especially since they looked at autologous epitopes for most of the participants and analyzed responses before and after the intervention. The PINCs are appropriate controls for the PICs.

D. The use of statistics is appropriate. The authors should discuss the Morisita-Horn Similarity Index (MHSI) used in Figure 4.

E. The conclusions are quite robust. It would be appropriate to have a limitations section in the discussion. One of the things they could mention is the fact that they predict epitopes to test rather than look at the whole proteome (which would take a lot of cells to do).

F1. It was not clear to me whether the differences in the surface proteome and gene signatures shown in figures 4h and 4i were seen in just PICs or also in PINCs. This should be clarified. If the latter, then the authors should discuss why this does not lead to improved CD8 stemness and post intervention control in these individuals

F2. I did not see a lot about the methods used for the TCR clonotypes in the methods section.

F3. I would like to see the pre and post intervention viral loads added to Table 1. If the table is too busy as is, a supplementary table with this information and perhaps the epitopes targeted should be included

F4. I think Extended Data Fig. 2e, should be included in the main figures, as the expression of gene signatures previously associated with follicular CD8+ T cells in lymphoid tissues is a very important finding.

F5. What percentage of total epitopes targeted were studied with tetramer analysis? The fact that the tetramer responses represented just a fraction of total responses should be acknowledged as a limitation.

G. Veenhuis RT et al JCI Insight 2018 should be cited as it showed CD8+ T cell mediated control of autologous virus in a post-treatment controller.

H. The manuscript is very well written. I would just like to see an acknowledgement of limitations in the discussion.

Referee #5

(Remarks to the Author)

Kiani et al. studied the characteristics of HIV-specific CD8 T cells before and after bNAb administration in 7 people who controlled viremia after ATI and bNAb washout and 5 who eventually rebounded. They showed that the post intervention controllers (PICs) exhibited HIV-specific CD8 T cell responses with higher stemness phenotype, proliferative capacity, and recall cytotoxicity compared to non controllers. These functional analyses were validated by single cell analyses showing signatures of stemness, metabolic fitness and reduced exhaustion in HIV-specific CD8 T cells from PICs. Interestingly, these functions were enhanced by bNAb administration in both groups. This higher functionality in PICs was not associated with any change in clonotype distribution, magnitude or breadth of responses. These data are suggesting a role for functional HIV-specific CD8 T cell responses in post bNAb control after ATI and support the development of strategies to enhance these functional responses in cure strategies.

The access to 4 different ATI trials is key for the work proposed as each ATI trial has only one or two PICs but this analysis allows for 7 PICs and 5 non controllers.

The analyses of HIV-specific CD8 T cells presented in this manuscript are exhaustive and a major strength of the work. The selection of matching class I HLA-optimal autologous peptides instead of consensus peptides commonly used is rarely seen but critical to generate reliable results. Each participant studied was epitope mapped and only the autologous responses were analyzed in this work.

The analyses were performed under conditions of undetectable viremia at both time points which prevents data to be confounded by viremia.

There are many interesting and novel observations presented in this manuscript:

The control of viremia post bNAbs was associated with HIV-specific CD8T cell proliferative capacity that was higher before ATI and further enhanced following the intervention.

The recall cytotoxicity was strongly associated with proliferative capacity.

Neither magnitude, breadth, nor induction of de novo responses were associated with control. These data also suggest that bNAb administration under aviremic conditions do not alter magnitude, breadth, nor induction of de novo responses.

Despite increased functionality and stemness, no change in clonotype frequencies were observed.

There are two comments to be addressed by the authors:

The participant from the eCLEAR study has a different sampling than the other participants as bNAbs were administered at time of ART initiation and responses were analyzed after bNAbs were gone pre-ATI. The data from this participant should be plotted differently or excluded.

The association between higher HIV-specific CD8 T cell stemness and post intervention control is not predictive as some participants in the PIC group have similar responses than in the non controller group and vice versa, so the word "predict" should be omitted as there is no causality demonstrated by the presented data.

Version 1:

Reviewer comments:

Referee #2

(Remarks to the Author)

This revised manuscript by Kiani et al. represents a clear improvement over the initial submission. The authors provide a detailed rebuttal that addresses several of the previously raised concerns, and they explain well some of the choices made, as well as the technical limitations (e.g., lack of samples) that make certain issues impractical—or even impossible—to resolve. The interpretations and limitations are now better articulated and aligned with the presented results.

Nevertheless, while this is an interesting paper and the findings are relevant to the field, concerns remain that limit the novelty and significance of the study. The reliance on a restricted panel of predefined epitopes remains a major limitation, as does the use of the clonotypic analysis strategy, which this reviewer respectfully disagrees with as being representative of the current state-of-the-art approach. While it is inherently challenging to demonstrate the absence of a phenomenon (here, the lack of generation of new responses and new TCR clones), this issue is central to the concept of a potential vaccinal effect of bNAb therapy (or its absence) and is not convincingly addressed in the current manuscript. The multiple comparisons of immune features, including stemness signatures, in an – understandably – small number of individuals raise the question of reproducibility in independent cohorts.

In addition, the challenges of achieving an integrated and consistent model across the cytometry and multi-omics datasets—particularly with respect to T-cell differentiation and clustering—remain unresolved.

Referee #4

(Remarks to the Author)

A. In this interesting manuscript, Kiani et al analyze the role of HIV-specific CD8+ T cell responses in postintervention control. They compare the responses in 7 post-intervention controllers (PICs) to those of 5 post-intervention non-controllers

(PINC)s). They measure breadth of responses to autologous epitopes, proliferation, stemness, TCR clonotypes and differential expression of genes and surface proteins using single cell RNA seq. They show convincingly that CD8+ T cell stemness correlates with the PIC outcome.

B. CD8+ T cell stemness has been associated with elite control, but PICs are a different subset of patients who do not have the same level of protective HLA alleles. The study is also important because it provides mechanistic insight into a phenomenon that was reported in the SIV model, but not often seen in human studies. Thus, this is an original and highly significant finding.

C. The methodology is quite robust; especially since they looked at autologous epitopes for most of the participants and analyzed responses before and after the intervention. The PINCs are appropriate controls for the PICs.

D. The use of statistics is appropriate. The authors have now adequately the one test I was not familiar with (Morisita-Horn Similarity Index).

E. The conclusions are very robust and the authors added the limitation section I asked for.

F. The authors addressed every single suggestion I made.

G. References are adequate, they added one I suggested.

H. The manuscripts really well.

Referee #5

(Remarks to the Author)

The authors have address all reviewers comments and I have no additional ones.

We sincerely thank the referees for their constructive assessments and suggestions. Our point-by-point responses appear below in blue italics and highlight corresponding revisions to the manuscript. These revisions, including new analyses, figures, and text, have strengthened the manuscript and clarified its conclusions.

Referee #1 (Remarks to the Author):

Kiani and colleagues compared T cell responses in patients with or without durable post-intervention control following administration of broadly neutralizing anti-HIV-1 antibodies. They observed a stronger T cell response in patients with durable control, in terms of magnitude but not breadth (as defined by the number of recruited clones). Further characterization revealed that these cells were already present in higher numbers prior to the intervention, showed a higher frequency of proliferation-competent progenitor or stem-like cells, and demonstrated superior effector capacity in ex vivo assays. Overall, the manuscript presents these findings in a robust and convincing manner.

What limits my enthusiasm in supporting the manuscript for publication in Nature is the absence of major conceptual novelty or a transformative breakthrough. While it is certainly valuable to report that such T cell populations can be identified and may be predictive of durable control, the findings are largely in line with what one might expect given the current understanding in the field. As such, I find it difficult to identify data within the manuscript that would, in my view, meet the threshold for Nature.

Overall, the reported conclusions are well supported, and the study is well executed in a challenging area of research.

My only remaining major concern is that the authors suggest the identified features could be used to stratify patients; however, they have not demonstrated the predictive power of these features in a larger, independent cohort. One reason for raising this point is that the observed differences in response magnitude, while statistically significant, appear relatively modest. This raises questions about whether they are sufficient for reliable prediction of clinical outcomes. Demonstrating such predictive capacity would substantially enhance the impact and translational relevance of the study.

We thank the referee for their positive assessment of the quality and relevance of the work, that the reported conclusions are well supported, and that the study was well executed. Despite including participants from four related interventional studies, the small number of participants achieving PIC provided insufficient statistical power to derive and validate a predictive mathematical model. Any implication of absolute predictive capacity was unintended, and we have therefore revised to avoid predictive language and have instead clarified that stemness precedes intervention (lines 2, 35, 87, 488). We have also made revisions to emphasize that prospective future trials will be required to formally evaluate predictive capacity (lines 246-248).

Referee #2 (Remarks to the Author):

General comments:

In this paper, the authors analyzed participants' samples and clinical data from four clinical trials involving infusion of the broadly neutralizing antibodies (bNAbs) 3BNC117 and 10-1074, followed by analytical treatment interruption (ATI). The study compared seven post-intervention controllers (PICs) and five post-intervention non-controllers (PINCs). Blood samples from pre- and post-intervention periods (6 or 12 weeks after ATI initiation, while viremia remained suppressed) were compared, focusing on HIV-specific CD8+ T cells. The authors hypothesize that by limiting the rate and magnitude of HIV recrudescence, bNAbs provide functional CD8+ T cell responses with a better opportunity to contain early viral rebound in lymphoid tissues. This mechanism may mediate PIC status after bNAb concentrations decline below therapeutic levels. They associate PIC with superior pre-intervention HIV-specific CD8+ T cell proliferative capacity, observed upon stimulation with autologous HIV peptides ex vivo, increased recall cytotoxicity against autologous HIV peptide-pulsed CD4+ T cells, enhanced HIV tetramer-positive CD8+ T cell stemness, improved metabolic fitness, and reduced T cell exhaustion. From their data, the authors also conclude that there is no emergence of new specificities or clonotypes of HIV-specific CD8+ T cells.

The study is relevant to the field, presents novel and interesting findings, and represents a considerable effort to gather data from several interventional trials. However, several issues limit the significance of the study and require further examination. There are also some overstatements and conclusions that are not sufficiently supported by the data. Addressing the specific comments below could significantly improve the manuscript.

We thank the referee for their positive assessment of the work as relevant, novel, and interesting, and for recognizing the level of effort involved. We also appreciate the referee's constructive suggestions for improvement. We address all specific concerns and highlight corresponding revisions to the manuscript below.

Specific major comments:

1. The experimental approach used to examine the patterns of epitopes targeted by HIV-specific CD8+ T cells pre- and post-bNAb administration is too narrow to support the sweeping conclusion that there is no emergence of new specificities or clonotypes. A notable strength of the study design is the use of autologous virus sequences. A weakness, however, is the use of limited panels of optimal epitopes matched to the HLA haplotypes of the participants. There are potential selection biases in such panels, which tend to preferentially include immunodominant epitopes elicited during infection in the absence of immunotherapy. Should bNAb administration lead to the emergence of new, less frequent specificities, they would be missed by this approach. The breadth of responses observed here—despite the use of autologous sequences (median of 3.5 epitopes)—suggests that detection is far from comprehensive, especially given that published studies using pools of overlapping peptides spanning the entire HIV proteome typically report a breadth of 10–30 CD8 epitopes, with around 15 specificities being common in ART-suppressed participants. Other studies on bNAb+ATI interventions suggest increased CD8+ T cell responses in both breadth (e.g., by ELISpot, as in this study) and magnitude (e.g., by ICS). Given how

critical these conclusions are for understanding possible mechanisms of control, the authors should explore this issue further. While PBMC numbers may be limiting in some participants, the authors should confirm their findings in at least a subset of participants using a more inclusive technique.

We thank the referee for raising this concern and for acknowledging the use of autologous epitope sequences as a strength of our work. We agree that our approach was not exhaustive nor did it definitively exclude a potential role for newly induced responses. Our approach did show that post-intervention control was not associated with detectable broadening of responses to known autologous HLA-optimal epitopes. We have made revisions to clarify this conclusion and to acknowledge the limitation that responses to as-yet undefined epitopes may have been missed by our approach.

Overlapping peptide (OLP) arrays spanning the entire HIV proteome include 769 consensus-sequence 15mer peptides, which would require 154 million PBMCs just for initial screening and more for validation. In addition to requiring more cells than we were able to obtain, this approach is prone to detection of class II-restricted CD4⁺ T cell responses and may miss responses against autologous epitope variants (inclusion of autologous variants would further increase the size and cost of such an approach). Limitations in specimen availability from these trials were a significant constraint; we carefully planned and executed our entire study using only 2-4 vials of 10-20 million PBMCs per sample. To maximize what could be learned from these precious samples and extend beyond what has already been examined in prior work, we focused on known and validated HLA class I-optimal epitopes, including all "A-list" HLA-optimal epitopes from LANL plus additional validated epitopes from IEDB, matched for participant HLA alleles and autologous provirus sequences. This also helped us to avoid studying escaped responses that may have misleading functionalities due to lack of chronic stimulation. While this approach may have missed responses against previously uncharacterized epitopes, it avoided a need for: (1) infeasible cell numbers for screening all OLPs, (2) subsequent validation of CD8⁺ T cell responses to HLA class I-optimal epitopes contained within targeted OLPs, and (3) subsequent evaluation of cross-recognition of autologous viral epitopes. Importantly, our approach enabled epitope-specific functional assays and the use of pHLA multimers for further investigation despite the limited material available, which was the primary intent of our study and would not have been feasible using an OLP approach.

*Niessl et al. (ref. 8) screened 123 overlapping clade B consensus sequence HIV-1 Gag peptides in 4 participants from one of the same trials included in our study, enabling some of the direct comparisons suggested by the referee. This work reported epitope-specific IFN- γ elispot breadths of 1, 1, 3, and 1 across the 123 Gag peptides screened at the aviremic pre-intervention time point. Consistent with this, across the 12 participants we studied, autologous HLA-optimal Gag epitope-specific breadths ranged from 0 to 3, whereas HIV-1 epitope-specific breadths to Gag and non-Gag epitopes ranged from 1 to 8. Two of the four participants studied by Niessl et al., 9255 (a PIC) and 9252 (a PINC), were also included in our study. Despite differences in thresholds and use of consensus OLPs versus autologous optimal peptides, the major results aligned: For PIC 9255, Niessl et al. detected 1 response out of 123 Gag OLPs screened. This response was to OLP5, which contains the A*03-restricted Gag RLRPGGKKK peptide that elicited the largest Gag-specific*

*response we observed from this participant. Notably, we also observed a large response to an autologous Pol variant that may have been missed using consensus OLPs. This was the only post-intervention controller included in the elispot analysis by Niessl et al., and emergence of new responses against Gag epitopes was not detected in this participant by either approach. For PINC 9252, we observed a response against B*39-restricted Gag TPQDLNTML, which is contained within OLP45, the target of the largest response reported in this participant by Niessl et al. Notably, we also observed a large response against an autologous Nef variant, which may have been missed using consensus OLPs. These data demonstrate the unique advantages of our approach and validate that we appear to have identified the major responses detected by an independent OLP screening approach, supporting our conclusion that post-intervention control is not associated with a broadening of responses against known epitopes.*

We have revised to clarify our conclusion (lines 31-33, 65-68, 73, 81-85, 162-163) and to discuss the limitations of our approach, including the possibility that we may have missed responses against previously uncharacterized epitopes or responses below the assay detection limit (lines 237-241). Importantly, this limitation does not invalidate or detract from the observed differences in functionality preceding intervention and the modest but statistically significant increase in functionality following intervention among detected responses, or from the main conclusions of our work.

2. Some important data and text are contradictory: While Figure 3c and Supplementary Fig. 2e support the claim that there is more pre-intervention stemness in PICs compared to PINCs, cluster 6 is one of the smallest and doesn't seem to reflect the flow data in Fig. 2h. Figure 3g shows upregulation of glycolysis pathways in PICs and enriched T cell cytotoxicity in PINCs, which contradicts the text. Supplementary Fig. 2e shows an increased frequency of exhausted T cells and Temra, not Tscm, in PICs post-intervention, again contradicting the text. Figure 4e contradicts the idea that PIC status is mediated by superior proliferation after bNAbs concentrations decline below therapeutic levels.

We thank the referee for bringing our attention to these apparent discrepancies, which we clarify here and in the revised manuscript. We have carefully re-evaluated our cluster annotation and conclusions to ensure appropriate caution when analyzing modest longitudinal changes.

Because flow cytometric assessment is based upon fewer parameters, multiomics captures additional heterogeneity. Thus, the pHLA⁺ CD45RA⁺ CD62L⁺ Tscm subset includes cells within other clusters beyond cluster 6, with sub-clustering driven by orthogonal features. Cells in cluster 4 co-express CD45RA and CD62L along with glycolytic signatures, so we have revised the annotation for cluster 4 as "Tscm/glyc" to better reflect this heterogeneity. The sum of frequencies of these two clusters more closely matches those observed by flow cytometry. However, differential cell numbers analyzed across responses may confound direct comparisons of absolute frequencies between flow cytometric and CITE-seq results, comparisons which we do not make in the manuscript. Our conclusion that the Tscm phenotype is elevated in PIC is supported by both data sets.

The glycolysis signature elevated in PICs shown in Fig. 3g was accurately described in the Results, but we have revised to remove any ambiguity (lines 149-155). The cytotoxicity-associated gene

sets upregulated in PINCs contain genes upregulated in activated T cells, consistent with our flow cytometric results in Fig. 4d and Fig. 4f, in which markers of activation and effector phenotypes are elevated in PINCs. These gene sets do not capture recall cytotoxicity, the potential for memory cells that are often in a resting or stem-like state *ex vivo* to mount recall responses and generate secondary cytotoxic effectors upon antigen re-exposure, which we directly measure. Unlike immediate cytotoxicity, which is frequently associated with exhausted T cells, recall cytotoxicity has been consistently associated with spontaneous HIV control and is also associated here with PIC. Therefore, these results are consistent with our conclusions.

Unlike pre-existing differences between groups, longitudinal changes in function and phenotype were an order of magnitude lower in all assays, including by CITE/TCR-seq. Of note, Fig. 4j (formerly Extended Data Fig. 2e) does show a modest increase in Tscm (cluster 6) following intervention, and this subset remains higher in PICs than in PINCs for both the pre- and post-intervention samples, consistent with our flow cytometry and functional data. However, any apparent inconsistencies between modest changes in subsets (including clusters 0 and 3) by flow cytometry vs multiomics are subject to the same limitations described above, and therefore we avoid conclusions based upon absolute or modest relative comparisons between these data sets and instead focus on consistent trends within each data set to support our conclusions. We have revised to ensure that modest longitudinal changes are interpreted with appropriate caution (lines 160-187).

Ki-67 staining in Fig. 4e indicates that *in vivo* proliferation is not detectable at statistically significant levels in peripheral blood at week 6/12 post-intervention. However, bNAb concentrations had not yet waned below therapeutic levels by this time point, which was selected intentionally to avoid confounding effects of viremia that subsequently arise in PINCs. Additionally, HIV recrudescence primarily originates in lymphoid tissues, and studies of PIC in nonhuman primates reported T cell activity detectable only in lymphoid tissues but not in peripheral circulation following bNAb administration. Therefore, it would be inappropriate to conclude that the absence of detectable proliferation in the periphery at a time point preceding waning of bNAb concentrations to subtherapeutic levels provides compelling evidence against our proposed model by which CD8⁺ T cell stemness may contribute to PIC. We have revised to acknowledge these limitations (lines 173-175, 245-246) and the need for additional studies including lymphoid tissue analyses designed to further address our proposed mechanistic hypothesis (lines 248-252).

3. Cluster annotation (Fig. 3, Extended Data Fig. 2) needs more supporting data: The study does not provide details on the gene lists and signatures used to annotate the clusters. The rationale for selecting the canonical markers for annotation is unclear. The authors should show the gene expression of canonical markers on the UMAP to better justify the cluster annotations. Clusters 0, 1, and 2 appear very similar on the heatmap, and the annotation requires further explanation. GZMB and CX3CR1 have been described in the context of T cell exhaustion (Tex); the authors' reasoning for annotating this cluster as TEMRA is not sufficiently explained. Similarly, for cluster 4, TCF7 is not expressed while TOX is highly expressed, which goes against the literature on Tpex. Plotting the top 10 DEGs for each cluster (can be

placed in the supplemental) would help. Once clusters are annotated with a phenotype, this should be used in the text, as readers do not remember cluster numbers.

We thank the referee for their assessment and suggestions. We have revised our Methods (lines 405-414) to detail our cluster annotation methods, including specific gene signatures displayed. Primary annotations were based upon CD45RA/RO and CD62L expression for general comparability to flow cytometric phenotyping and we have revised our annotations to provide secondary designations based upon differential expression results to better reflect their heterogeneity. We have revised to display key markers supporting our annotations using bubble plots to indicate fractions of cells expressing each marker (Fig. 3d). As suggested, we have revised to provide additional supporting evidence for cluster annotations, including feature plots projected onto UMAP plots and bubble plots for the top genes and surface markers upregulated in each cluster (Extended Data Fig. 2). Temra cells are typically characterized by CD45RA⁺/RO⁻ CD62L⁻ and expression of terminally differentiated effector cell signatures, consistent with cluster 3, which is low for canonical exhaustion markers. Although cluster 4 expresses TCF7 comparably to cluster 2 (Tcm), we have revised to annotate cluster 4 as Tscm/glyc to reflect that this cluster co-expresses CD45RA and CD62L but also expresses glycolytic gene signatures and shares features with proliferating transitory cells derived from stem-like precursors (lines 150-152), distinguishing this subset from resting Tscm in cluster 6 and Tex cells in cluster 0. We have also revised to refer to cluster annotations instead of numbers in the text following their introduction, as suggested.

4. The authors analyzed TCR clonality using the Morisita-Horn index to compare the similarity of pre- and post-intervention TCR clones (Figure 4). They conclude that there are no new epitopes in PICs post-intervention. While the high Morisita-Horn index values indeed do not suggest a major reshaping of the CD8⁺ T cell repertoire post-treatment, this should be more rigorously assessed. By visual inspection alone, it is not possible to determine the number of clones, and this should be formally tested. The Simpson diversity index could be used to measure the diversity within a single timepoint. Moreover, it is well established in the field that examining only the beta chain does not accurately describe a TCR clone. The authors should also use the alpha chain to perform clonal analysis. Using both CDR3 α and CDR3 β sequences to define a TCR clone, as opposed to using only CDR3 β , reduces the likelihood of falsely grouping TCR clones that happen to share the same beta chain. This approach captures a higher level of TCR diversity, as using only beta chains can lead to an overestimation of clonal expansion when distinct clones share the same beta chain.

We thank the referee for their suggestions. We have revised the manuscript to clarify our conclusion that diversification or emergence of major clonotypes within the analyzed responses following intervention were not uniquely associated with PIC (lines 164-166). Consistent with our functional results, longitudinal changes were modest and comparable between PIC and PINC groups with no evidence of clonotypic emergence unique to PICs. We have also revised to add Simpson diversity indices and the number of unique clones for each response and sample, as suggested, to Supplementary Table 3. The Morisita-Horn index is related to the Simpson diversity index but is generally insensitive to differences in sample size, making it a more rigorous metric for comparative assessments of population diversity than comparisons of Simpson diversity of numbers of unique clones.

We respectfully disagree that CDR3 β sequences define TCR clones less accurately than CDR3 α/β pairs in practice. Independent rearrangement of CDR3 β s with identical junctional diversity in multiple distinct clones within the same response from the same participant is exceedingly rare, whereas allelic inclusion (~20% of T cells express two TCR α alleles) and low detection rates of CDR3 α relative to CDR3 β from mRNA frequently result in overestimated diversity when assigned based upon CDR3 α/β pairs. For these reasons, CDR3 β alone is frequently used as a basis for clonality analyses to avoid separation or discarding of cells without detected CDR3 α or those expressing either of two CDR3 α alleles within the same clonal population.

Nevertheless, to address this referee's concern, we have reanalyzed using CDR3 α/β pair-based assignment for comparison, shown below (y-axes). The results are highly concordant with CDR3 β -based assignments (x-axes) and do not alter our conclusions. Most of the inflated diversity in CDR3 α/β -based assignment was attributable to CDR3 β clones being split into two CDR3 α/β clones in which one had detectable CDR3 α and the other did not, but these are highly unlikely to represent biologically distinct clones. The remaining minority of discrepant clones represented two different CDR3 α/β pairs sharing the same CDR3 β , but these were within the expected range of reported CDR3 α allelic inclusion frequencies. There were no instances in which any CDR3 β was paired to more than two CDR3 α across the data set. Our conclusion that diversification or emergence of major clonotypes within the analyzed responses following intervention were not uniquely associated with PIC remained supported by both analyses with high concordance. As the CDR3 β -based clonotype assignments appear to more accurately reflect meaningful biology, we elected to report these in the manuscript. Although this methodological comparison is tangential to our manuscript's conclusions, we are happy to include it in the extended data if the editor and referees feel it is important to do so.

5. More generally, and in line with comment #1: Given the small study groups (which is an expected limitation), the authors should be more nuanced when interpreting the lack of statistically significant differences between PIC and PINC for some of the parameters measured as proof of similarity or absence of effect of the bNAb+ ATI intervention.

We appreciate this comment and have revised to ensure appropriate nuance for all conclusions, given limitations in sample size, as highlighted above. We indeed report several modest changes induced by intervention in both groups, although none were uniquely associated with control, unlike the tenfold larger differences in functionality observed prior to intervention.

Specific minor comments:

Figure 1:

- In addition to Table 1, the authors should provide a comprehensive table listing the sequence and number of successfully tested peptides for each donor.
- They should also indicate in Figures 1 and 2 which peptide elicited which response, and in which donor.
- Please check the figure legend for Fig. 1c—the numbers do not seem to add up.

We have revised to include the suggested response-level data (Supplementary Table 1, second tab), to use participant-specific symbols throughout the manuscript figures with a corresponding key in Table 1, and to plot Fig. 1c with horizontal jittering such that data points no longer overlap.

Figure 2:

- Please add CFSE histogram overlays of undivided and proliferative peaks for representative PIC and PINC samples in the supplementary material.
- Please indicate the IDs of the two PINC donors with high recall cytotoxicity but low proliferation, and comment on why these donors may behave differently.
- The authors state that recall cytotoxicity was strongly associated with proliferative capacity, supporting a role for highly functional HIV-specific CD8⁺ T cells in PICs. Was proliferative capacity also correlated with the IFN γ response?
- Given that PINCs hardly expand in response to peptide stimulation, it is surprising that the 5-day expansion yielded enough cells to perform the killing assays.

We have revised to add the requested CFSE histogram overlays (Supplementary Fig. 1a), to add representative examples from both groups in the main figure (Fig. 2b), and to indicate all participants throughout each figure in the manuscript using unique symbol-color combinations, with a key added to Table 1. The presence of functional CD8⁺ T cell responses in a minority of PINCs may suggest that they lack additional, as-yet undefined features required for PIC and/or may have developed bNAb resistance resulting in rapid virus recrudescence and ART re-initiation despite CD8⁺ T cell stemness. We have revised to avoid predictive language and to highlight the importance of measuring additional immune parameters in future studies to further evaluate this (lines 250-252). Consistent with prior literature, IFN- γ magnitude was weakly correlated with proliferation ($r=0.47$) and cytotoxicity ($r=0.33$) in our data, whereas proliferation and cytotoxicity were strongly correlated with one another ($r=0.80$). We have revised to clarify that proliferation is a better correlate of cytotoxic potential than IFN- γ production (lines 88-89).

Use of tetramers and dextramers:

- Please provide a comprehensive list of the specific tetramers used for each donor, and explain the validation process.
- Were the same pHLA monomers used to generate the tetramers and dextramers?
- Please provide additional evidence to distinguish the Tscm population in Fig. 2g from naïve T cells (Tn).
- The data show that PICs have a higher frequency of HIV- and CMV-tetramer+ cells compared to PINCs at both timepoints (Fig. 4c). The authors should comment on the potential significance of this observation.
- The perforin staining shown in Supplementary Fig. 1c is not convincing—the “negative” population already appears quite bright.

We have revised to list the tetramers used (Supplementary Table 1, second tab) and to describe their validation in Methods (lines 340-343). Dextramers were not used or mentioned in this study, but we have revised to clarify that the same monomers were used for tetramers in cytometry and multiomics (lines 361-363). Tscm cells in Fig. 2g stained brightly for pHLA tetramer, whereas naïve T cells should not bind to pHLA tetramers at measurable frequencies; clonal expansion among Tscm is also evident in Extended Data Fig. 3d. As differences in pHLA⁺ frequencies in Fig. 4c were not statistically significant between or within any groups, we elected not to speculate about their potential biological relevance. We have revised to display a CMV-specific response from the same experiment that more clearly demonstrates separation and gating of perforin staining (Fig. S1c).

scRNA-seq and CITE-seq on sorted HIV-specific CD8⁺ T cells:

In addition to the major comments already raised, please also address the following:

- Provide the staining profiles of the dextramer-sorted cells to validate the sorting strategy.
- Explain how the sample matrix was structured and how batch effects were corrected for in the scRNA-seq and CITE-seq analyses.
- Clarify the discrepancy in the number of PIC and PINC cells analyzed. Provide the number of cells per donor and per dextramer that were sequenced.
- Explain the metric used to integrate pathway analysis into the heatmap (e.g., z-score).
- Describe the statistical approaches used for contrast and pathway analyses, and discuss why the single-cell data resolution was not better integrated into these analyses.
- Clarify the observation that the CMV response is mainly driven by a TCM-like cluster, which is not apparent in the flow cytometry data in Extended Fig. 1d.
- For Fig. 4b, explain the rationale for showing only the CDR3b sequences of 6 out of 9 PIC dextramers and 4 out of 5 CMV dextramers. Clarify the number of epitope responses observed in each group (PIC, PINC, CMV), and reconcile any discrepancies with the information provided in Extended Fig. 2d.
- Can the authors clarify what Tpm refers to?

Representative barcoded tetramer staining profiles and sorting gates are shown in Fig. 3a. We have revised the Methods section to clarify that we included an HLA-mismatched individual to exclude nonspecific binding of barcoded tetramers and ensure sorting of bona fide antigen-specific cells (lines 365-367). We excluded cells without detection of pHLA barcodes or with TCRs observed only once across the data set from downstream analyses to further exclude potential nonspecific cells, as described in Methods (lines 385-389, 419-420). Batch correction was not applicable because the entire 10X data set was generated from a single pool of barcoded cells as

a single sequencing library and run, which we have revised to clarify in Methods (lines 368-375). We sorted as many cells as feasible for CITE-seq, limited by specimen availability and response size, rather than normalizing to the least abundant responses. Sorted and analyzed cell numbers for each response, sample, and participant are now reported (Supplementary Table 1, third tab). Additional metadata and the full data set are available via GEO:GSE294440 (access token kfalmuakttwdxcb), which will be publicly released at the time of manuscript publication.

We have revised the Methods to provide more detail regarding statistical analyses of our 10X data, including that gene signatures were integrated using the AUCell metric (lines 401-417). Single-cell resolution was fully integrated into each analysis, including analyses of differential mRNA, surface protein, pathways, and TCR clonality between clusters, between phenotypes, between specificities, and between longitudinal samples. We have replaced the heatmap in Fig. 3d with a bubble plot to better communicate single-cell resolution, including detection rates.

Despite enrichment for CD62L⁺CD45RA⁻/RO⁺ cells expressing several Tcm-associated markers, cluster 2 is heterogeneous and includes cells of other phenotypes, including Tem, offering a potential explanation for the difference between flow cytometric and multiomics annotations of CMV-specific cells. Such direct comparisons between cytometry and annotated multiomics also suffer from the limitations described above, including differential sampling between responses. Importantly, this apparent minor discrepancy does not affect any of our conclusions.

Longitudinal analyses in Fig. 4 include all responses from which longitudinal samples were analyzed, which is further clarified in the revised Methods and figure legend (lines 416-417, 531-533). Data from all responses with greater than 10 cells, including those without longitudinal sampling, are reported in Supplementary Table 3 and Extended Data Fig. 3d. We have revised the annotation of cluster 7 to “CD127⁺CD73⁺” and included citations to its previously reported stemness (lines 180-183, refs. 41-42).

Referee #3 (Remarks to the Author):

I co-reviewed this manuscript with one of the reviewers who provided the listed reports.

Referee #4 (Remarks to the Author):

A. In this interesting manuscript, Kiani et al analyze the role of HIV-specific CD8+ T cell responses in post-intervention control. They compare the responses in 7 post-intervention controllers (PICs) to those of 5 post-intervention non-controllers (PINCs). They measure breadth of responses to autologous epitopes, proliferation, stemness, TCR clonotypes and differential expression of genes and surface proteins using single cell RNA seq. They show convincingly that CD8+ T cell stemness correlates with the PIC outcome.

B. CD8+ T cell stemness has been associated with elite control, but PICs are a different subset of patients who do not have the same level of protective HLA alleles. The study is also important because it provides mechanistic insight into a phenomenon that was reported in the SIV model, but not often seen in human studies. Thus, this is an original and highly significant finding.

C. The methodology is quite robust; especially since they looked at autologous epitopes for most of the participants and analyzed responses before and after the intervention. The PINCs are appropriate controls for the PICs.

We thank the referee for their accurate summary of the major findings, for appreciating the novelty and significance of the work, and for complimenting the robustness of the study design.

D. The use of statistics is appropriate. The authors should discuss the Morisita-Horn Similarity Index (MHSI) used in Figure 4.

We have revised to clarify that MHSI quantifies overlap/diversity of clonotypic distributions between two populations while accounting for differences in sample size (lines 423-425). Initially defined in ecology studies, MHSI measures the similarity of two populations and is interpreted as the likelihood that individuals (cells) drawn at random from each population will be of the same species (clonotype), relative to the likelihood of drawing the same species (clonotype) twice from either population. This metric is commonly used in TCR analyses as it is more robust to differences in sample size than comparing Simpson diversity or the number of unique clones.

E. The conclusions are quite robust. It would be appropriate to have a limitations section in the discussion. One of the things they could mention is the fact that they predict epitopes to test rather than look at the whole proteome (which would take a lot of cells to do).

We appreciate this suggestion and have revised to include a limitations section in the Discussion noting this and other limitations (lines 236-252).

F1. It was not clear to me whether the differences in the surface proteome and gene signatures shown in figures 4h and 4i were seen in just PICs or also in PINCs. This should be clarified. If the latter, then the authors should discuss why this does not lead to improved CD8 stemness and post intervention control in these individuals

We observed that both PICs and PINCs exhibited modest but statistically significant increases in proliferative functionality following intervention (Fig. 2c). Due to limited statistical power and given that modest effects were observed in both groups, we performed longitudinal analyses (Fig. 4) on samples from all participants for whom longitudinal sampling was available to maximize statistical power, as noted in the revised Methods and figure legend (lines 416-417, 531-533). Most changes in Figs. 4g-i were apparent in both groups, including the oxidative phosphorylation gene signatures in Fig. 4i, consistent with our functional assays. However, pre-intervention differences in functionality/stemness were already tenfold higher in PICs, whereas longitudinal changes were less than two-fold in both groups (Fig. 2c, 4g-h). Our interpretation is that the pre-existing differences in functionality contribute much more to PIC than the modest changes following intervention, which occur in both groups but were insufficient to elicit control of viremia in PINCs. We have revised to further clarify this interpretation in the manuscript (including lines 183-187).

F2. I did not see a lot about the methods used for the TCR clonotypes in the methods section.

We have revised to describe these methods in greater detail (lines 419-426).

F3. I would like to see the pre and post intervention viral loads added to Table 1. If the table is too busy as is, a supplementary table with this information and perhaps the epitopes targeted should be included

Viral loads were undetectable at both the pre and post-intervention time points studied. We have revised to clearly state this both in Methods and in the Table 1 legend (lines 274-275, 473-474) and have also added a supplementary table reporting the epitopes targeted (Supplementary Table 1).

F4. I think Extended Data Fig. 2e, should be included in the main figures, as the expression of gene signatures previously associated with follicular CD8⁺ T cells in lymphoid tissues is a very important finding.

We have revised to include this panel in the main article (now Fig. 4j) and to include additional support for differential lymphoid CD8⁺ T cell-associated gene expression between clusters and their enrichment in Tscm and CD127⁺CD73⁺ cells (Fig. 4k).

F5. What percentage of total epitopes targeted were studied with tetramer analysis? The fact that the tetramer responses represented just a fraction of total responses should be acknowledged as a limitation.

We agree and have revised to list the analyzed responses/multimers (Supplementary Table 1) and to more explicitly acknowledge this limitation (lines 241-243). Tetramer analysis covered 16 (33%) of 48 detected responses, weighted toward the most abundant responses in each participant to the extent feasible. Importantly, our proliferation analysis included all detected responses.

G. Veenhuis RT et al JCI Insight 2018 should be cited as it showed CD8⁺ T cell mediated control of autologous virus in a post-treatment controller.

We agree and have revised to include this citation in the revised manuscript, as well as some related literature (lines 207-208, 228-231).

H. The manuscript is very well written. I would just like to see an acknowledgement of limitations in the discussion.

We agree this improves the manuscript and have revised to discuss key limitations (lines 236-252).

Referee #5 (Remarks to the Author):

Kiani et al. studied the characteristics of HIV-specific CD8 T cells before and after bNAb administration in 7 people who controlled viremia after ATI and bNAb washout and 5 who eventually rebounded. They showed that the post intervention controllers (PICs) exhibited HIV-specific CD8 T cell responses with higher stemness phenotype, proliferative capacity, and recall cytotoxicity compared to non controllers. These functional analyses were validated by single cell analyses showing signatures of stemness, metabolic fitness and reduced exhaustion in HIV-specific CD8 T cells from PICs. Interestingly, these functions were enhanced by bNAb administration in both groups. This higher functionality in PICs was not associated with any change in clonotype distribution, magnitude or breadth of responses. These data are suggesting a role for functional HIV-specific CD8 T cell responses in post bNAb control after ATI and support the development of strategies to enhance these functional responses in cure strategies.

The access to 4 different ATI trials is key for the work proposed as each ATI trial has only one or two PICs but this analysis allows for 7 PICs and 5 non controllers.

The analyses of HIV-specific CD8 T cells presented in this manuscript are exhaustive and a major strength of the work. The selection of matching class I HLA-optimal autologous peptides instead of consensus peptides commonly used is rarely seen but critical to generate reliable results. Each participant studied was epitope mapped and only the autologous responses were analyzed in this work. The analyses were performed under conditions of undetectable viremia at both time points which prevents data to be confounded by viremia.

There are many interesting and novel observations presented in this manuscript:

The control of viremia post bNAbs was associated with HIV-specific CD8T cell proliferative capacity that was higher before ATI and further enhanced following the intervention.

The recall cytotoxicity was strongly associated with proliferative capacity.

Neither magnitude, breadth, nor induction of de novo responses were associated with control. These data also suggest that bNAb administration under aviremic conditions do not alter magnitude, breadth, nor induction of de novo responses.

Despite increased functionality and stemness, no change in clonotype frequencies were observed.

We thank the referee for their thorough review, their appreciation of the unique strengths of this work, and for their enthusiasm regarding the significance of its conclusions.

There are two comments to be addressed by the authors:

The participant from the eCLEAR study has a different sampling than the other participants as bNAbs were administered at time of ART initiation and responses were analyzed after bNAbs were gone pre-ATI. The data from this participant should be plotted differently or excluded.

We have revised to clearly distinguish eCLEAR participant 107 (and all individual participants) throughout the manuscript by using distinct symbols for each participant, with a key in Table 1. We have also revised to more explicitly acknowledge how this sample was handled (lines 272-274). Because only a post-intervention sample from eCLEAR 107 was included, it was excluded from all analyses involving pre-intervention samples and from all longitudinal analyses. Therefore, its inclusion did not impact any of the main conclusions, which rely primarily upon pre-intervention

and longitudinal comparisons. Importantly, all unpaired post-intervention PIC vs post-intervention PINC comparisons and correlative analyses were robust to the exclusion of data from eCLEAR participant 107, which was intermediate and did not drive any conclusions. Thus, we decided to include it, as this participant is of interest to many in the field, but to ensure transparency about this limitation, which is now clearer in the revised manuscript.

The association between higher HIV-specific CD8 T cell stemness and post intervention control is not predictive as some participants in the PIC group have similar responses than in the non controller group and vice versa, so the word “predict” should be omitted as there is no causality demonstrated by the presented data.

We have revised to omit the word “predict” (by replacing it with “precede”, lines 2, 35, 87, 488) and to more clearly highlight the importance of evaluating predictive capacity prospectively in future trials (lines 246-248).

Response to Referees:

We thank the referees for their feedback. Our point-by-point responses appear below in blue text.

Referee #2 (Remarks to the Author):

This revised manuscript by Kiani et al. represents a clear improvement over the initial submission. The authors provide a detailed rebuttal that addresses several of the previously raised concerns, and they explain well some of the choices made, as well as the technical limitations (e.g., lack of samples) that make certain issues impractical—or even impossible—to resolve. The interpretations and limitations are now better articulated and aligned with the presented results.

Nevertheless, while this is an interesting paper and the findings are relevant to the field, concerns remain that limit the novelty and significance of the study. The reliance on a restricted panel of predefined epitopes remains a major limitation, as does the use of the clonotypic analysis strategy, which this reviewer respectfully disagrees with as being representative of the current state-of-the-art approach. While it is inherently challenging to demonstrate the absence of a phenomenon (here, the lack of generation of new responses and new TCR clones), this issue is central to the concept of a potential vaccinal effect of bNAbs therapy (or its absence) and is not convincingly addressed in the current manuscript. The multiple comparisons of immune features, including stemness signatures, in an – understandably – small number of individuals raise the question of reproducibility in independent cohorts.

In addition, the challenges of achieving an integrated and consistent model across the cytometry and multi-omics datasets—particularly with respect to T-cell differentiation and clustering—remain unresolved.

We thank the referee for their assessments and are pleased that they found the revised manuscript to be clearly improved. While we acknowledge the study's limitations, we wish to reiterate that our manuscript does not conclude the absence of a vaccinal effect. Some new responses and clonotypes were observed at low frequencies, and modest increases in stemness/proliferative capacity were consistently evident across responses; both of these findings are consistent with previous reports of a potential vaccinal effect. However, these were not distinguishing features of PICs relative to PINCs, which was not possible to determine in prior studies due to the small number of PICs present in each individual study. We also wish to reiterate that our primary findings were consistent across data sets despite differences in sampling and methodology. While we acknowledge the inherent sample size limitations of this study, our primary conclusion that CD8⁺ T cell stemness and proliferative capacity precede post-intervention control is already being replicated in independent cohorts by other groups, as discussed in the manuscript.

Referee #4 (Remarks to the Author):

A. In this interesting manuscript, Kiani et al analyze the role of HIV-specific CD8⁺ T cell responses in postintervention control. They compare the responses in 7 post-intervention controllers (PICs) to those of 5 post-intervention non-controllers (PINCs). They measure breadth of responses to autologous epitopes, proliferation, stemness, TCR clonotypes and differential expression of genes and surface

proteins using single cell RNA seq. They show convincingly that CD8+ T cell stemness correlates with the PIC outcome.

B. CD8+ T cell stemness has been associated with elite control, but PICs are a different subset of patients who do not have the same level of protective HLA alleles. The study is also important because it provides mechanistic insight into a phenomenon that was reported in the SIV model, but not often seen in human studies. Thus, this is an original and highly significant finding.

C. The methodology is quite robust; especially since they looked at autologous epitopes for most of the participants and analyzed responses before and after the intervention. The PINCs are appropriate controls for the PICs.

D. The use of statistics is appropriate. The authors have now adequately the one test I was not familiar with (Morisita-Horn Similarity Index).

E. The conclusions are very robust and the authors added the limitation section I asked for.

F. The authors addressed every single suggestion I made.

G. References are adequate, they added one I suggested.

H. The manuscripts really well.

We thank the referee for their assessments and are pleased that they found the revised manuscript to satisfactorily address all of their feedback.

Referee #5 (Remarks to the Author):

The authors have address all reviewers comments and I have no additional ones.

We thank the referee for their assessments and are pleased that they found the revised manuscript to satisfactorily address all referees' comments.